# A surface lipoprotein on *Pasteurella multocida* binds complement factor I to promote immune evasion

Quynh Huong Nguyen, Chun Heng Royce Lai, Michael J. Norris, Dixon Ng, Megha Shah, Christine Chieh-Lin Lai, David E. Isenman, Trevor F. Moraes *

Department of Biochemistry, University of Toronto, Toronto, Ontario, Canada

* trevor.moraes@utoronto.ca

## Abstract

*Pasteurella multocida* is the leading cause of wound infections in humans following animals' bites or scratches. This bacterium is also commonly found in the respiratory tract of many mammals and can cause serious diseases resulting in the rapid death of infected animals, especially cattle. To prevent these infections in cattle, a subunit-based vaccine utilizing the surface lipoprotein PmSLP was developed and showed remarkable protection with a single dose administration. Here, we report that PmSLP binds host complement factor I (FI) and facilitates cleavage of complement components C3b and C4b independently of any cofactors (e.g., FH, C4BP), thereby allowing the pathogen to evade host defence. Cryo-EM structure of PmSLP bound to FI reveals that PmSLP stimulates FI enzymatic activity by stabilizing the catalytic domain. This is the first time that a bacterial protein has been shown to directly activate FI independent of complement cofactors and target all arms of the complement cascade.

## Author summary

*Pasteurella multocida*, a zoonotic pathogen, causes serious diseases in many animal species. This bacterial infection has a tremendous economic impact on the farming industry, especially for small holder farmers. Outbreaks of *P. multocida* infection are often found in cattle populations resulting in bovine respiratory diseases and hemorrhagic septicemia, both of which devastating diseases. A recently developed protein-based vaccine showed remarkable protection against *P. multocida*. The main component of this vaccine is a surface lipoprotein, PmSLP, found on the surface of *P. multocida*. We identified PmSLP as a complement factor I-binding protein. Our findings also revealed that PmSLP could stimulate the proteolytic activity of factor I, which is a crucial mechanism to inhibit the complement cascade and increase the chance of bacterial survival inside the host.

**Data availability statement:** All data supporting the findings of the current study are available within the paper and its Supplementary Information or Source data files. Coordinates and structures of PmSLP-1 and PmSLP-1:FI complex have been deposited in the Protein Data Bank (PDB) with accession codes 9B3E and 9B3H, respectively. The associated maps of PmSLP-1:FI complex have been deposited in the Electron Microscopy Data Bank under the accession code EMD-44139 (monomeric assembly of PmSLP-1:FI), EMD-44146 (conformation 1 of the dimeric assembly of PmSLP-1:FI complex), and EMD-44149 (conformation 2 of the dimeric assembly of PmSLP-1:FI complex). Source data are provided with this paper.

**Funding:** 1. A portion of this research was supported by NIH grant U24GM129547 and performed at the PNCC at OHSU and accessed through EMSL (grid.436923.9), a DOE Office of Science User Facility sponsored by the Office of Biological and Environmental Research. This research was supported by equipment purchased in part by the CFI and operating funds provided by the NSERC to TFM (RGPIN-2018-06546). QHN is supported by the Ontario Graduate Scholarship (OGS) and NSERC CGS-M and PGS-D. The funders had no role in study design, data collection and analysis, decision to publish, or preparation of the manuscript.

**Competing interests:** I have read the journal's policy and the authors of this manuscript have the following competing interests: TFM and CCLL, are co-authors on a patent, "Slam polynucleotides and polypeptides and uses thereof" - Patent Number: WO2017136947A1. TFM is a co-author on a provisional patent, "Veterinary vaccines and methods for the treatment of Pasteurella multocida infections in food production animals" - United States Provisional Application No. 63/332,966 or WO/2023/201434 and PCT/CA2023/050537.

## Introduction

*Pasteurella multocida* is a Gram-negative, zoonotic bacterium commonly found in the upper respiratory tract of various animals. Under stressful conditions, this commensal microbe can become pathogenic, causing serious symptoms that manifest as various diseases in their hosts [1,2]. Although *P. multocida* has been isolated from a wide range of hosts, it is frequently found to infect farm animals, especially cattle. It has been estimated that 50% of annual cattle deaths in North America could be attributed to *P. multocida*-induced pneumonia (or bovine respiratory disease - BRD) [2]. Moreover, certain bovine isolates, mostly from Asia and Africa, cause hemorrhagic septicemia (HS), which incurs an annual loss of roughly $800 million USD [3]. Due to the rapid clinical onsets of *P. multocida* infection, infected animals often succumb to death within days of disease development. Although the mechanism of infection remains unclear, *P. multocida* has been found to spread from the site of infection to other organs [4–6]. Thus, to travel through the bloodstream, the pathogen must possess the ability to evade complement-mediated killing.

The complement system plays an important role in host defence as it provides rapid and effective clearance of invading microbes. Complement activation leads to the deposition of C3b and C4b, or opsonin molecules, on the cell surface. These molecules serve as signals for phagocytosis and trigger the formation of the membrane attack complex on the cell membrane. Stationed at virtually every step of the cascade are regulatory proteins whose purposes are to preclude spontaneous complement activation and prevent healthy host tissues from collateral damage. These negative regulators have specificity for host cells, permitting complement inactivation only on the host cell surface [7,8]. To overcome this tight surveillance system, bacterial pathogens have evolved several strategies to increase their survival rates, and one such mechanism involves recruiting complement regulatory proteins to the bacterial surface [9]. Pathogens notorious for their virulence, such as *Neisseria meningitidis* and *Staphylococcus aureus*, are found to possess the ability to shut down the complement cascade by acquiring complement regulatory proteins. The two commonly identified factors captured by these pathogens are factor H (FH) and C4b-binding protein (C4BP) [9–14]. At physiological conditions, C4BP binds preferentially to C4b, while FH prefers C3b. The binding of C4BP and/or FH to the opsonin molecules then allows factor I to cleave C3b and C4b into their inactive form and inhibit all 3 branches (classical, lectin, and alternative pathways) of the complement cascade [15]. Thus, this mechanism of hijacking the host's complement regulatory proteins provides bacterial pathogens with the same protection that would otherwise be exclusive to host cells.

In *Neisseria meningitidis*, the protein responsible for capturing complement factor H belongs to a specific group of outer membrane proteins known as Slam-dependent surface lipoproteins (SLP), specifically FH binding protein [12,16]. Slam is the bacterial Type XI secretion system protein responsible for delivering lipoproteins across the bacterial outer membrane. Many Slam homologs have been identified throughout the gamma proteobacteria phylum, and several Slam substrates, including those from

clinically relevant pathogens, were found to be virulence factors [16,17]. Hooda *et al* discovered the presence of Slam and its putative lipoprotein substrate in *P. multocida,* termed PmSLP*.* We showed that Slam can potentiate the surface display of PmSLP in lab strains of *E. coli*, suggesting that it is likely the case in *P. multocida* as well [17]. However, the functional implications of the display of PmSLP on the *P. multocida* surface remain to be explored.

In *P. multocida*, there are various virulence factors, such as capsule, toxins, and outer membrane proteins, that are often targeted for vaccine development [18–20]. In our recent study, we showed that PmSLP, could elicit strong immune responses against bovine isolates. We also found that the gene *pmSLP* was highly prevalent amongst *P. multocida* strains isolated from cattle. To date, we have identified four PmSLP variants that share roughly 30–40% amino acid sequence identity to each other [21]. Interestingly, the segregation of these variants appears to correlate with the distribution of bacterial serogroup and disease specificity. For instance, PmSLP-1 and -2 are found in serogroup A strains associated with BRD, whilst PmSLP-3 is generally linked to serogroup B or HS-causing strains. The variant, PmSLP-4, is an exception as it has also been isolated from a broad range of hosts. Although PmSLP is present in 97% of all bovine isolates, the low sequence identity and lack of cross-reactivity between the variants suggest that PmSLP homologs are structurally, and perhaps functionally, different. Regardless, when used as vaccine immunogens, these surface antigens have been shown to provide protection against invasive disease caused by *P. multocida* in both mice and cattle [21]. Thus, studies aimed at characterizing these surface lipoproteins could provide new insights into their roles in pathogenesis and disease manifestation.

This present study focuses on the first variant, PmSLP-1, which is present in 85.5% of bacterial strains that cause bovine respiratory diseases [21]. We show that PmSLP-1 not only exhibits a high-affinity interaction with bovine complement factor I, but it also serves as a co-factor for factor I to promote cleavage of complement C3b and C4b. This protein-protein interaction thus plays a crucial role in modulating the complement activation on the bacterial cell surface and increasing bacterial resistance to complement-mediated killing. While several bacterial proteins capable of binding to factor I have been identified, not much is known about the molecular details of their interactions with factor I. To our knowledge, this study provides the first high-resolution structure of factor I in complex with one such bacterial protein and identifies a novel immune evasion strategy in *P. multocida*.

## Results

### PmSLP-1 exhibits characteristic features of a Slam-dependent surface lipoprotein

All *P. multocida* strains harbouring a *pmSLP* gene share a common genetic arrangement where a putative *pmSLP* gene is found downstream of a putative *slam* gene. We previously isolated a Slam and PmSLP pair from *P. multocida* strain 70, an avian isolate, and showed that Slam is responsible for the translocation of this PmSLP across the outer membrane [17]. To confirm that this is also the case in bovine isolates, we cloned out the Slam-PmSLP-1 operon from *P. multocida* strain 36950 and assessed the surface display of PmSLP-1 in a lab strain of *E. coli*. We transformed *E. coli* C43 cells with plasmid encoding either PmSLP-1 or Slam+PmSLP-1. To avoid cellular toxicity from protein over-expression, we relied on the basal expression of the pET52 vector system to produce our proteins of interest. We then performed a proteinase K shaving assay on the bacterial cells to check for the surface display of PmSLP-1 (Fig 1A). In the absence of proteinase K, we observed strong signals for PmSLP-1 regardless of the presence of Slam. However, when treated with proteinase K, PmSLP-1 co-expressed with Slam became susceptible to cleavage activity, as seen in the loss of FLAG-tag signal (Fig 1B). In agreement with our previous study, the proteinase K shaving assay indicates that Slam is needed for successful translocation of PmSLP-1 across the outer membrane.

To identify structural features of PmSLP-1 that would aid in its function, we sought a high-resolution crystal structure of the protein. Previous attempts at crystallizing the protein using the PmSLP-1[15] construct yielded no success. Thus, to improve the chance of protein crystallization, we modified the protein construct by removing the first 94 N-terminal residues, predicted to be disordered, and mutating a stretch of surface-exposed, charged residues (E315A, K316A, and

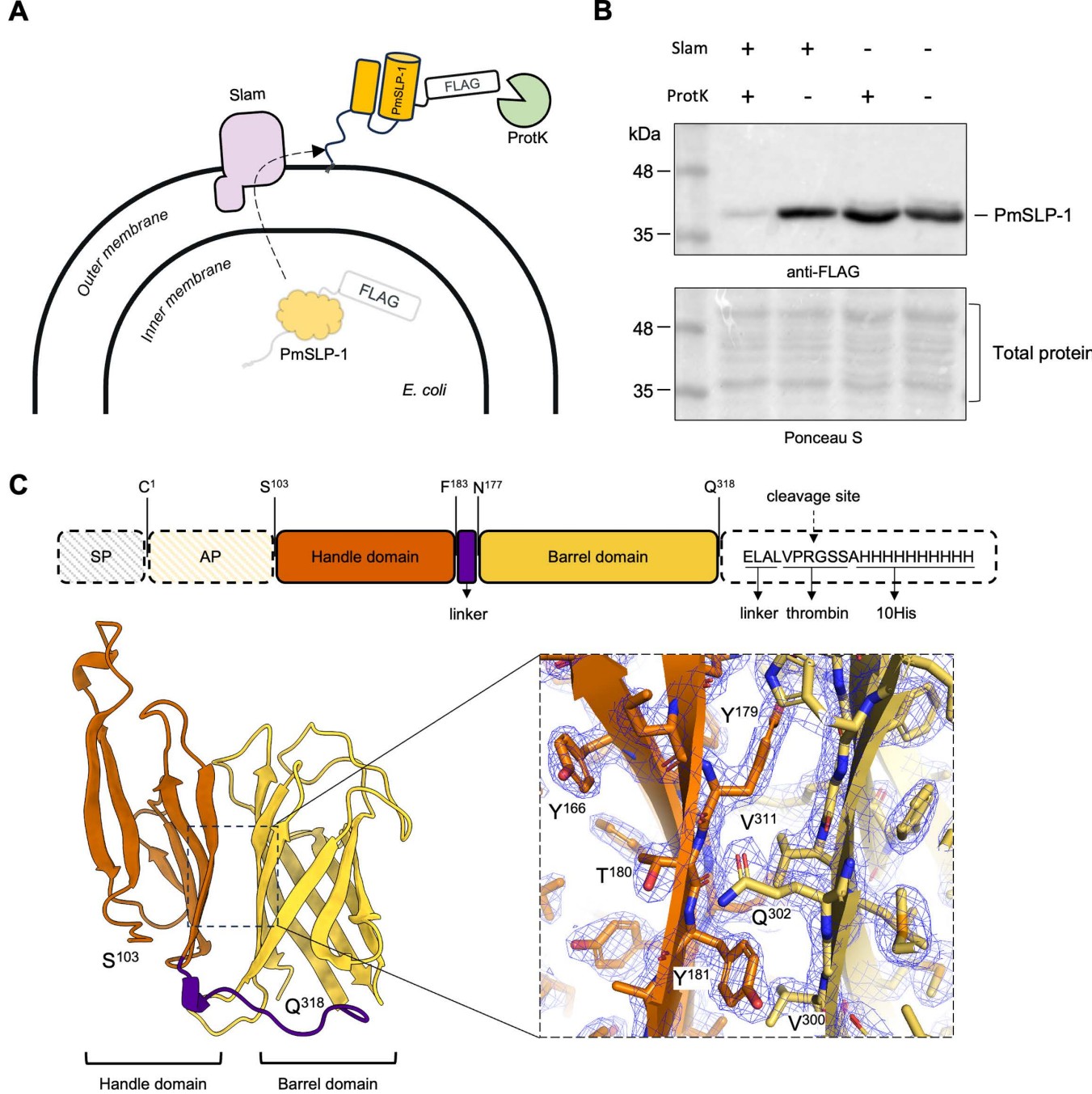

**Fig 1. PmSLP-1 exhibits characteristic features of a Slam-dependent surface lipoprotein.** (A) A model showing the proposed translocation of PmSLP-1 by Slam to the surface of *E. coli.* Surface exposed PmSLP-1 would be susceptible to cleavage by proteinase K. (B) Representative Western blot analysis for the proteinase K shaving assay. *E. coli* expressing either PmSLP-1 with Slam or PmSLP-1 alone were treated with proteinase K. Presence of PmSLP-1 remained after proteinase K treatment was detected using anti-FLAG antibodies (top panel). Proteins were stained with ponceau S to show the amount of sample loaded per lane and detect cell lysis (bottom panel). n = 3 independent experiments. (C) Schematic representation of the domain organization and the solved crystal structure of PmSLP-1 at 2.0 Å. The signal peptide (SP) and anchoring peptide (AP) were excluded in the construct used for structural study. The His tag was removed by thrombin cleavage during protein purification process. The inset shows the electron density at the interface between the handle domain and the barrel domain (contour level = 1.5 sigma).

K317A) at the C-terminus [22,23]. By removing the flexible N-terminal region and lowering the surface entropy on the C-terminus, we were able obtain protein crystals with high diffraction quality. The modified variant PmSLP-1[95] and its selenomethionine derivative were then used to solve and refine the protein structure to a resolution of 2.0 Å (Fig 1C and S1 Table). PmSLP-1 is a predominantly β-stranded structure which forms an N-terminal domain (NTD) and a C-terminal domain (CTD) connected by a 13-residue linker peptide (Fig 1C). The eight β-strands on the C-terminus fold into a barrel, coined the barrel domain, and the six β-strands on the N-terminus form an open-hand-like structure, referred to as the handle domain. These two domains are held stably together through a network of hydrogen bonds (S2 Table). We also observed two short helices present in the loop regions of the protein. Noticeably, the overall architecture of PmSLP-1 resembles that of previously known Slam-dependent SLPs (S1 Fig) [17,24]. We postulate that the conserved β-barrel CTD serves as a Slam-recognition motif whereas the varying NTD and loop regions dictate the protein function through specific protein interactions.

## PmSLP-1 binds complement factor I from ruminants

To understand the role of PmSLP-1 in *P. multocida*, we sought to identify its binding partner through affinity purification coupled with mass spectrometry. Purified FLAG-fusion PmSLP-1 protein was immobilized onto anti-FLAG resins and incubated with bovine serum. Proteins enriched by the affinity purification process were trypsinized and analyzed by liquid chromatography-tandem mass spectrometry (LC-MS/MS). The mass spectrometry data identified factor I as the mammalian interactor of PmSLP-1 (S3 Table). To account for non-specific interactions, we also performed pull-down experiments with empty FLAG resin and a FLAG-tagged TbpB – a transferrin binding protein from *Neisseria gonorrhoeae*. We used TbpB as a negative control because it is also a Slam-dependent SLP, and it interacts specifically with human transferrin. The presence of FI was exclusive to the PmSLP-1 sample, which was confirmed through Western blot analysis (Fig 2A).

Additionally, since *P. multocida* is a zoonotic pathogen, we wondered whether PmSLP-1 could interact with FI from other mammalian hosts. Following the co-immunoprecipitation experimental protocol described above, we incubated purified FLAG-tagged PmSLP-1 with several normal animal sera. Pulldown results were analyzed on SDS-PAGE, and the presence of FI was detected by Western blot analysis. We found that PmSLP-1 interacted with FI from bovine, sheep, and goat sera, indicating a preference for ruminant host complement factors (Fig 2B). Further analysis of the protein sequences of FI homologues reveals that FI from bovine (*B. taurus*), sheep (*O. aries*), and goat (*C. hircus*) share more than 90% sequence similarity and are more closely related to each other than to FI from other species (S2 Fig). These observations suggest that the binding of PmSLP-1 to FI is not restricted to bovine hosts, but it is specific to ruminants.

## PmSLP-1 forms a high-affinity complex with bovine factor I in solution

Next, we investigated whether PmSLP-1 and bovine FI could form a stable complex in solution. We first purified PmSLP-1 and FI individually and subjected the proteins to an analytical size-exclusion chromatography assay (S3A Fig). The overlay elution profiles of FI and PmSLP-1 show distinct peaks, with retention volumes of 13.2 mL (peak 2) and 14.5 mL (peak 3), respectively. Upon mixing the purified proteins together, with PmSLP-1 present in excess, we observed a clear shift towards a higher molecular weight for all FI populations (peak 1), suggesting a formation of the protein complex. The presence of both species in peak 1 was verified by SDS-PAGE analysis (S3A and S3B Fig).

We then proceeded to evaluate the binding affinity between PmSLP-1 and bovine FI. To do so, we immobilized biotinylated bovine FI onto streptavidin biosensors and performed binding analysis using biolayer interferometry (Fig 2C-F). We evaluated the binding of FI to two PmSLP-1 constructs: PmSLP-1[15] and its truncated version, PmSLP-1[95]. As mentioned, the crystal structure of PmSLP-1 was obtained with PmSLP-1[95], which has a significant portion of the anchoring peptide removed (Fig 1C). While this region is predicted to be disordered, it is not known whether this N-terminal region is important for protein function. The kinetic profiles for the binding of bovine FI to both PmSLP-1 variants indicate high affinity binding regardless of the presence of the N-terminal disorder region (Fig 2C, 2D and 2F). Similarly, using steady state

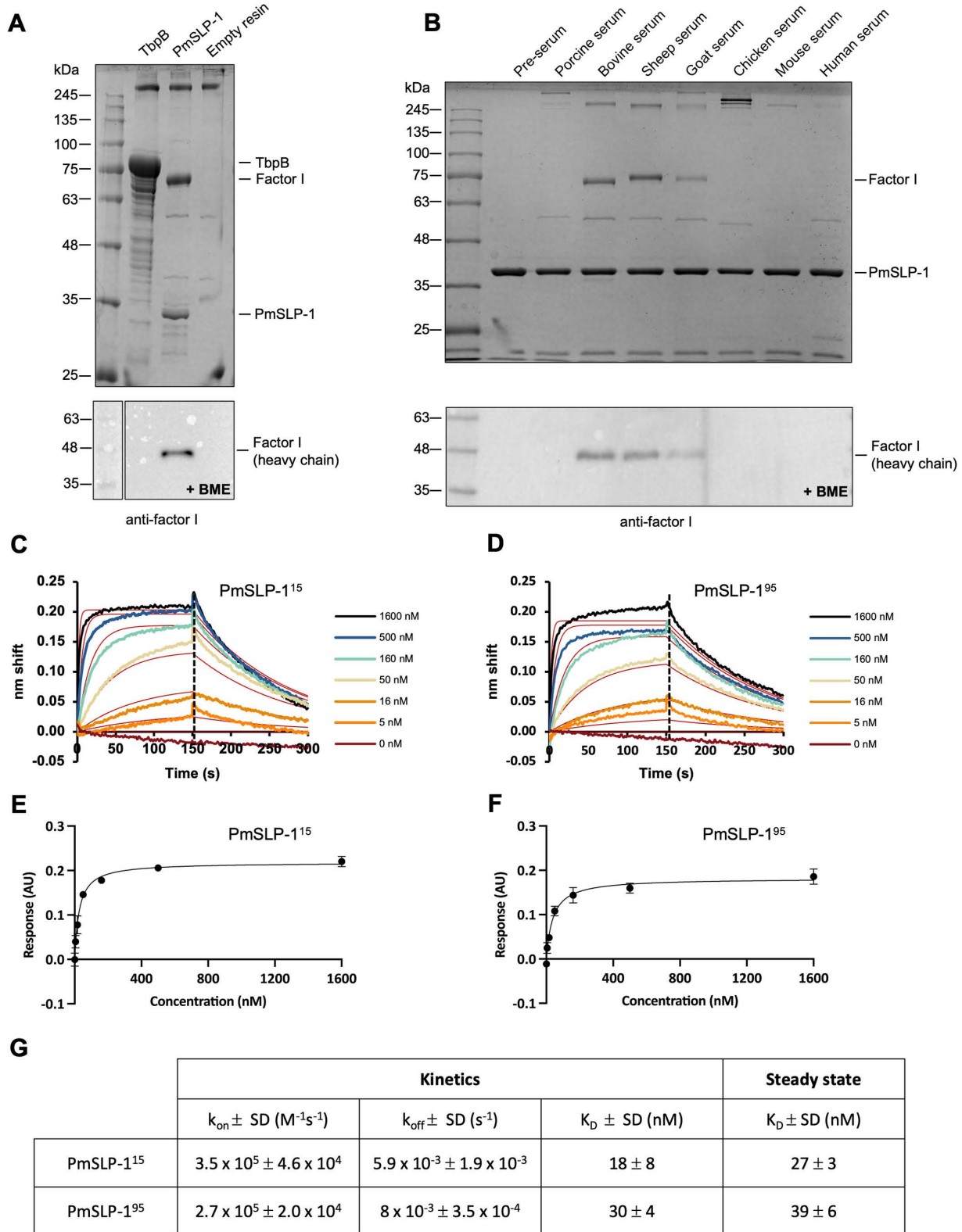

**Fig 2. PmSLP-1 binds and forms stable complex with bovine complement factor I.** (A) A representative SDS-PAGE analysis of the serum pull-down assay showing the interaction between PmSLP-1 and bovine FI. Transferrin binding protein B (TbpB) and empty FLAG resin were used as

| | Kinetics | | | Steady state |
|---|---|---|---|---|
| | $k_{on} \pm$ SD ($M^{-1}s^{-1}$) | $k_{off} \pm$ SD ($s^{-1}$) | $K_D \pm$ SD (nM) | $K_D \pm$ SD (nM) |
| PmSLP-1[15] | $3.5 \times 10^5 \pm 4.6 \times 10^4$ | $5.9 \times 10^{-3} \pm 1.9 \times 10^{-3}$ | $18 \pm 8$ | $27 \pm 3$ |
| PmSLP-1[95] | $2.7 \times 10^5 \pm 2.0 \times 10^4$ | $8 \times 10^{-3} \pm 3.5 \times 10^{-4}$ | $30 \pm 4$ | $39 \pm 6$ |

negative controls. Samples were analyzed under non-reducing conditions (top panel). Western blot analysis of the pull-down samples under reducing conditions. The heavy chain of bovine FI was detected with anti-human FI antibodies (bottom panel). (B) A representative SDS-PAGE analysis of the pull-down experiment where PmSLP-1 was incubated with various animal sera; samples were analyzed under non-reducing conditions (top panel). Western blot analysis of the pull-down samples under reducing conditions (bottom panel). The heavy chain of bovine FI was detected with anti-human FI antibodies. (C-F) Representative biolayer interferometry sensorgrams and saturation curves showing binding of biotinylated bovine FI to PmSLP-1$^{15}$ and PmSLP-1$^{95}$. PmSLP-1 was present at various concentration as indicated in the sensorgrams (solid-coloured lines), and a 1:1 binding model was used to fit the association and dissociation data (red). The saturation curves were derived from the steady state values of the corresponding sensorgrams. (G) Binding kinetics and steady state binding constants for biotinylated bovine FI with wild type PmSLP-1$^{15}$ and PmSLP-1$^{95}$. The $k_{on}$, $k_{off}$, and binding constants represent three independent experiments and are presented as mean±SD.

analysis, we found that PmSLP-1$^{95}$ bound to FI with a dissociation constant ($K_D$) of 39±6 nM, which is comparable to the binding affinity between PmSLP-1$^{15}$ and FI ($K_D = 27±3$ nM) (Fig 2E-G and 2F). This suggests that the removal of the disordered N-terminal domain does not impact the protein function. Additionally, the thermal shift assay results revealed no significant difference in the inflection temperatures ($T_0$ and $T_1$) of PmSLP-1$^{15}$ and PmSLP-1$^{95}$, indicating that the N-terminal truncation does not affect either the ternary structure or the stability of the protein (S3C and S3D Fig).

## PmSLP-1:FI structure determination

To obtain more insight into the interaction between PmSLP-1 and FI, we performed single-particle cryo-electron microscopy (cryo-EM) on the purified complex (S4A and S4B Fig). An initial cryo-EM dataset suggested that the particles exhibit preferred orientation. Thus, to address this issue, we collected roughly 2,300 movies at 40° tilt, which enabled an initial 3D reconstruction of the complex at a resolution of 3.9 Å [25]. Additional processing allows us to identify two populations of the complex, both with C2 symmetry (Fig 3A). We used AlphaFold2 to generate a starting model of PmSLP-1:FI complex; both proteins contain long disordered N-terminal regions, which were trimmed off due to low model confidence score (S4C Fig) [26]. For each map, we were able to fit two copies of the predicted structure simply by rigid body fitting of the entire protein complex (S4D Fig). The maps suggest that PmSLP-1:FI complex exists as a dimer, with the dimeric interface forged between two FI molecules (Figs 3A and (S4D). By overlaying the two maps and aligning the models on protomer 1, we observed a 17° or a 5 Å difference in the relative positions of protomer 2 in the two populations (S4D Fig). While the complex dimerization was an interesting observation, no heterogenous binding was detected in our BLI experiment between bovine FI and PmSLP-1 in solution (Fig 2C). Additionally, previous studies on the structure of FI in solution indicate that FI remains a monomer even at a concentration that is higher than its serum concentration [27,28]. Thus, we suspect that the dimerization of the complex might be an artifact of the sample freezing process for cryo-EM, and it is unlikely that the dimerization would occur *in vivo*.

We then performed symmetry expansion on the particles from each of the dimeric complex and local refinement to improve the resolution of each protomer, which resulted in a final EM map exhibiting an overall resolution of 3.5 Å (Figs 3A and S5 and S4 Table). The final refined model of PmSLP-1:FI complex differs slightly from the AlphaFold2 predicted model (RMSD = 0.81 Å) with most of the variation occurring in the PmSLP-1 structure (S5B Fig). When compared to the unbound crystal structure of PmSLP-1, we noted a loss of the short helix in the handle domain of PmSLP-1 in the bound state but no major conformational changes (Fig 3B, top left panel). Since the estimated local resolution of this region in the cryo-EM map was quite low (> 4.5 Å), we were unable to confidently build and resolve the secondary structure feature here. However, this loop situates at the complex interface, so it is also possible that binding to FI results in this structural change. Analysis of the PmSLP-1:FI structure shows that PmSLP-1 forms contacts with the light chain of FI, with 1374 Å$^2$ of interface area calculated from PISA analysis (Fig 3B) [29,30]. The electrostatic potential maps reveal clusters of charged residues and small hydrophobic patches on the interacting surfaces of both proteins (Fig 3C). Specifically, hydrophobic residues W417, L418, and Y499 of bovine FI interact with a hydrophobic patch on PmSLP-1, formed by A116, A120, I128, L129, and I175 (Fig 3B, bottom left panel). A second hydrophobic cluster consists of F422 on FI making

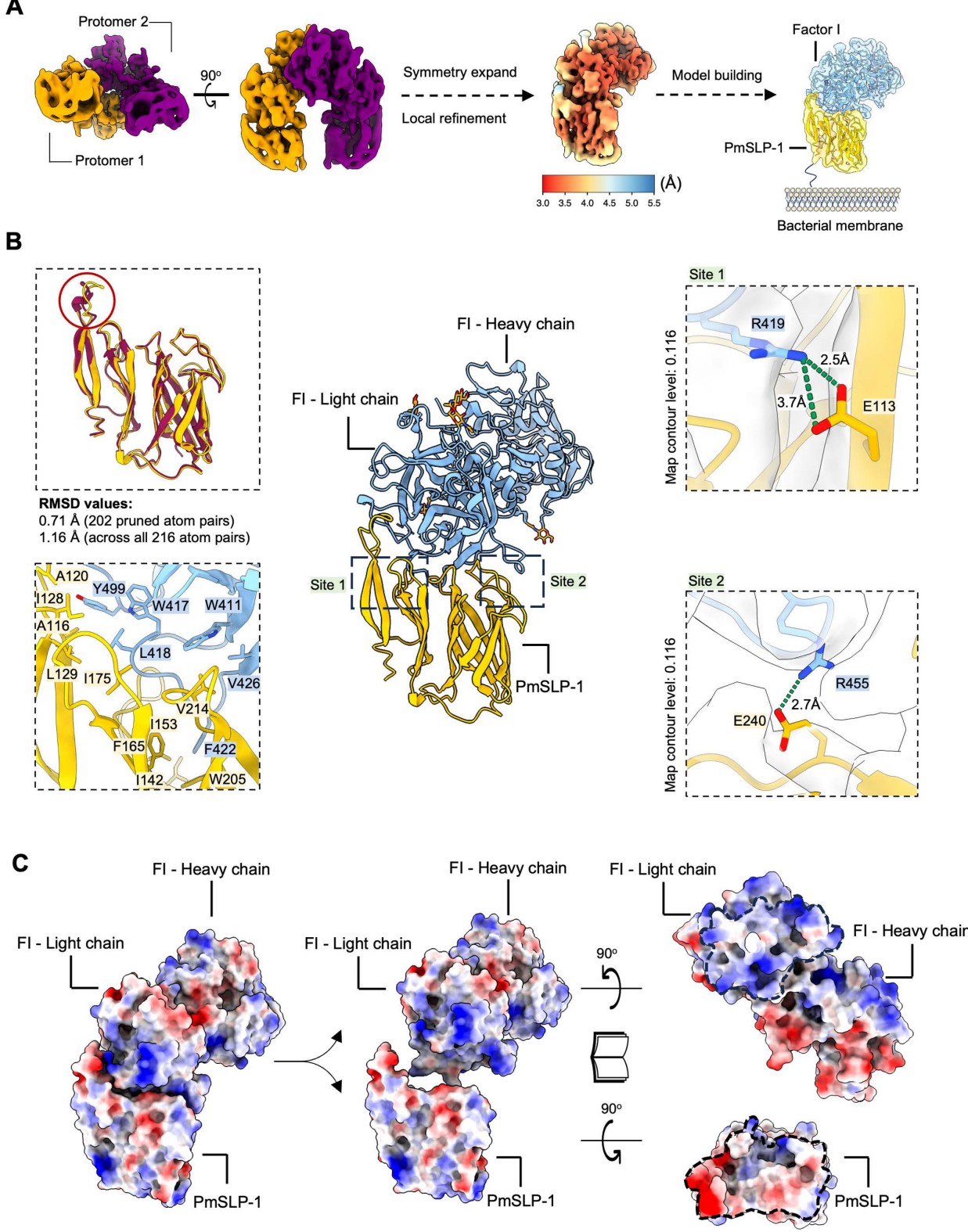

**Fig 3. Cryo-EM structure of PmSLP-1 in complex with bovine factor I.** (A) Cryo-EM map of PmSLP-1 in complex with bovine FI (top and side views), suggesting a dimer of the protein complex (left). The protomers are shown in different colours. The density of the complex monomer was

obtained via symmetry expansion and local refinement; the estimated local resolution of the map is shown (middle). The atomic model of the protein complex is docked into the cryo-EM density map (right), with PmSLP-1 shown in yellow and bovine FI shown in blue. The color scheme is maintained throughout. (B) Atomic model of PmSLP-1:FI complex in cartoon representation. The top left inset shows an overlay between PmSLP-1 from the cryo-EM complex (in yellow) structure and the solved crystal structure of PmSLP-1 (in dark purple); the red circle highlights the short helix that is missing in the cryo-EM structure. The bottom left inset depicts the hydrophobic contacts between PmSLP-1 and FI. The right insets showcase the residues that form salt bridges between PmSLP-1 and FI. (C) Electrostatic potential maps at the interface between PmSLP-1 and bovine FI with the interface footprint outlined; red = negative, blue = positive.

contacts with I142, I153, F165, and W205 of PmSLP-1 (Fig 3B, bottom left panel). And finally, V214 of PmSLP-1 interacts with W411 and V426 of FI (Fig 3B, bottom left panel). In addition, this protein-protein interaction is supported by two salt bridges and a network of H-bonds at the interface (Fig 3B, right panels and S5 Table). Notably, the residues on bovine FI that form salt bridges with PmSLP-1 are conserved only in sheep and goat (consistent with the pulldown results), which further support the host specificity of this interaction (S6 Fig).

Although PmSLP-1 and FH do not share structural homology, they both bind to the same region on FI (S7A Fig, left panel) [31]. However, unlike FH, PmSLP-1 binds FI with high affinity even in the absence of C3b. To gain a better understanding of the interactions between FI with either FH or PmSLP-1, we analyzed these protein complexes using PISA analysis software (part of the CC4 package) [29,30]. Further analysis of the interface statistics suggests that the interaction between PmSLP-1 and FI is comparable to, if not stronger than, that of FH and FI (S7A Fig, right panel). Given that PmSLP-1 and FH recognize the same surface on FI, we wondered whether the arrangement of PmSLP-1 and FI in the cryo-EM complex would allow access for the substrate, e.g., C3b. We first used the crystal structure of FH:FI:C3b as a guide to model the PmSLP-1:FI:C3b complex (S7B and S8A Figs). While the small size of PmSLP-1 allows it to fit in the groove on C3b where FH sits, we observe some clashes between the C-terminal barrel domain of PmSLP-1 and the MG6 domain of C3b (S8A Fig, left inset). However, the relative position of FI and C3b in this predicted complex model would still allow FI to access and cleave the first scissile bond on C3b (S8A Fig, right inset). To resolve the issue of the clashes between PmSLP-1 and the MG6 domain of C3b, we performed an AlphaFold3 prediction for PmSLP-1, bovine FI, and bovine C3b [32]. The general arrangement of each component in the predicted complex is consistent with our docked model and the crystal structure of FH:FI:C3b (S8B Fig). Additionally, no clash is observed between PmSLP-1 and C3b, and the interacting surface between PmSLP-1 and bovine FI is accurately predicted (S8B Fig, left inset). However, unlike its predecessor, i.e., AF2, AlphaFold3 was unable to correctly model the handle domain of PmSLP-1 (S8B Fig, right inset). When using FI as an anchor to align the AF3 model with our manually docked model, we noted differences in the position of the MG rings and the CUB-TEB domain relative to the CTC domain. While the structure of C3b itself is not altered, this shift in the position of these domains completely removes the clash between PmSLP-1 and C3b. Therefore, by combining the AF3 model and our cryo-EM structure, we were able to gain some insights into how the proteins might orient themselves in a ternary complex to facilitate the FI-mediated processing of C3b. With this PmSLP-1:FI:C3b model, we also revisited the observed cryo-EM maps of PmSLP-1:FI dimers. S8C Fig revealed significant clashes between the CTC domain of C3b of one protomer with FI of the other promoter, which means the dimerization of FI would prevent access to C3b and is functionally inefficient. Taken together, while PmSLP-1 and FH bind the same ligand, FI, PmSLP-1 likely possesses distinct mechanisms to activate FI and influence downstream cleavage of the substrates, C3b and C4b.

Since we could not resolve the interface to high resolution, we used cross-linking mass spectrometry (XL-MS) to further validate the binding interface of the PmSLP-1:FI complex. The purified complex sample was treated with a disuccinimidyl suberate (DSS) cross-linker, and the cross-linked samples were separated by non-reducing SDS-PAGE. We observed multiple bands at roughly 36 kDa, 68 kDa, and 120 kDa which would correspond to PmSLP-1, FI, and PmSLP-1:FI complex, respectively (S9A Fig). The additional bands at higher molecular weight could be attributed to crosslinking events between protein complexes. For the XL-MS experiment, only the 110 kDa band was isolated, subjected to in-gel tryptic digestion, and analyzed with LC-MS/MS. We identified 39 unique crosslinked peptides, including 32 intra-protein (17 for

**A**

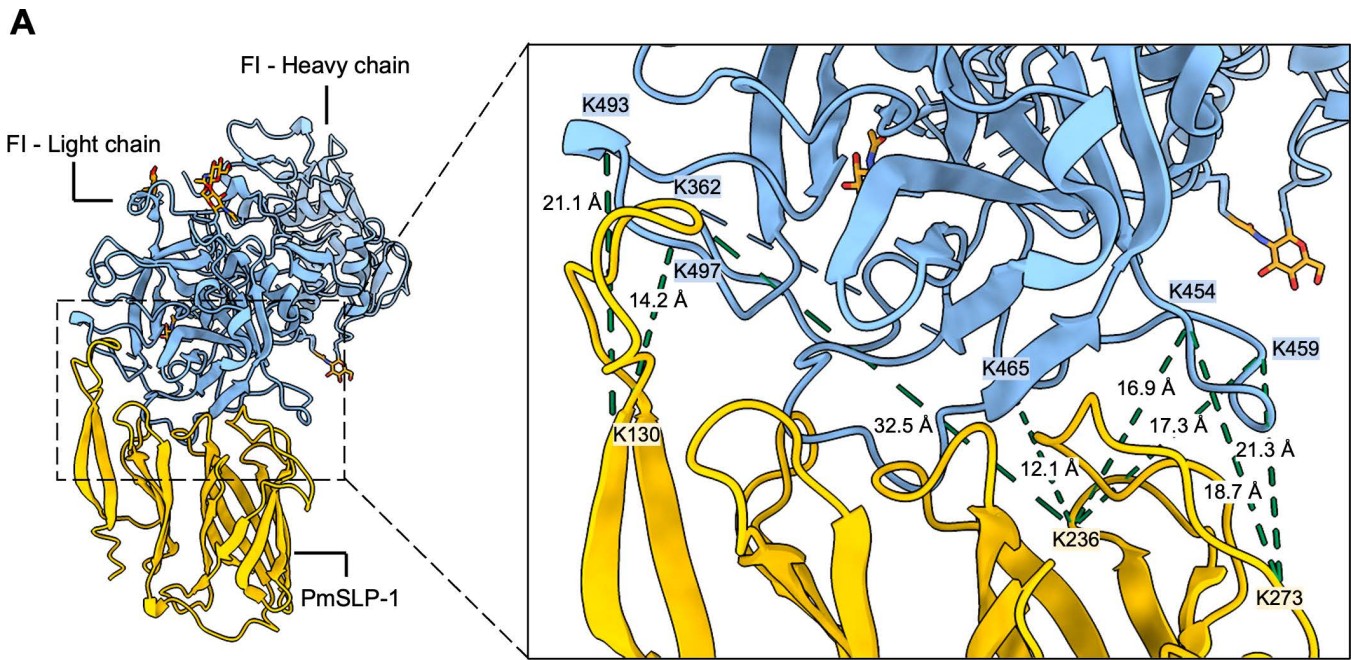

**B**

| PmSLP-1 variants | $K_D \pm$ SD (nM) | P value |
|---|---|---|
| Wildtype | $27 \pm 3$ | N/A |
| V214D | *No binding detected* | |
| E240A | *No binding detected* | |
| D210A | $272 \pm 46$ | *** (p = 0.0008) |
| D121A | $117 \pm 11$ | *** (p = 0.0002) |
| R167A | $107 \pm 21$ | ** (p = 0.0031) |
| D124R | $53 \pm 6$ | ** (p = 0.005) |
| L127D | $85 \pm 27$ | * (p = 0.021) |
| I133D | $43 \pm 11$ | NS (p = 0.091) |
| K236E | $32 \pm 7$ | NS (p = 0.363) |
| E132R | $28 \pm 15$ | NS (p = 0.962) |

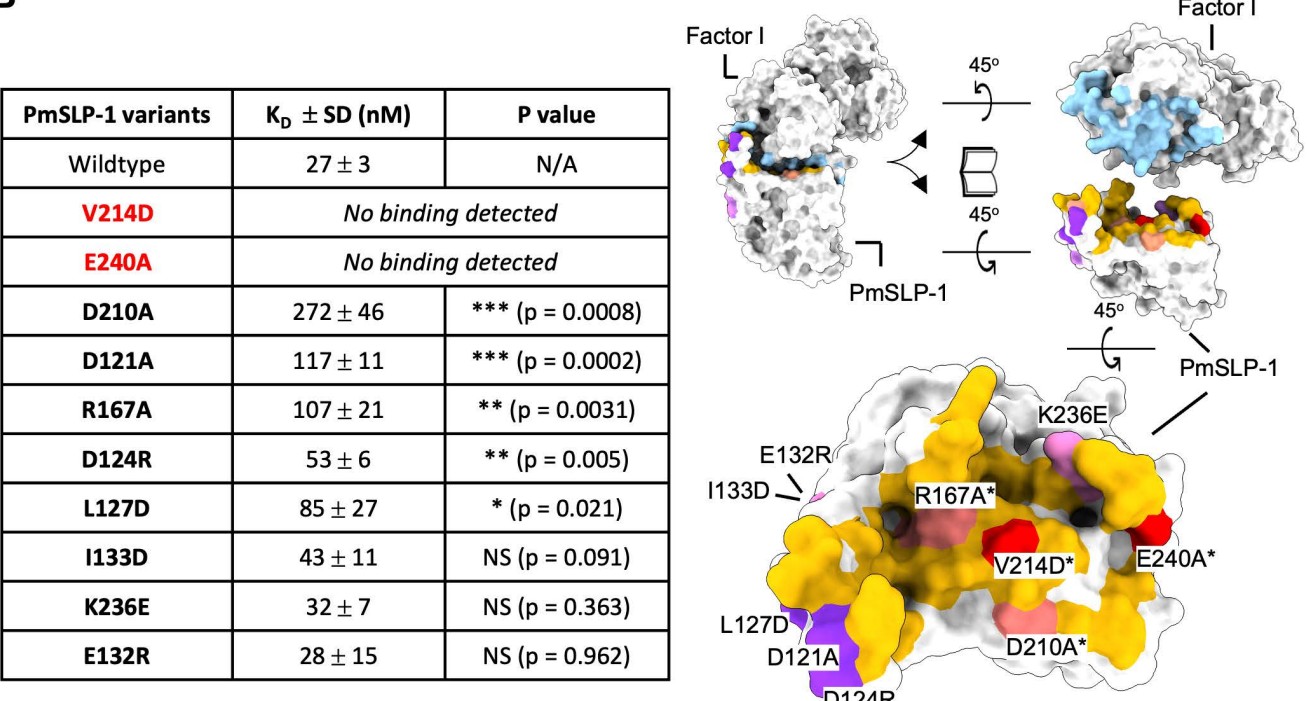

**Fig 4. Validating the interface between PmSLP-1 and bovine factor I.** (A) Inter-protein cross-links (dotted lines) are mapped on the atomic model of the protein complex. The cross-linked lysine residues and the $C_\alpha$-$C_\alpha$ distance between each lysine pairs are indicated. (B) Ten solvent-exposed residues on PmSLP-1 were mutated and assessed for their effects on the binding of PmSLP-1 to bovine FI. The dissociation constant ($K_D$) for each mutant was obtained via biolayer interferometry assay (n = 3) and compared with the $K_D$ of the wild-type PmSLP-1[15]. NS = not statistically significant. The mutated residues are then mapped on the structure of the protein complex. The cryo-EM structure of PmSLP-1:FI is shown as white surface. The interfacing residues on PmSLP-1 and FI, identified through PISA analysis, are coloured yellow and light blue, respectively. The residues on PmSLP-1 selected for mutagenesis studies are labelled in the lower right panel. The residues annotated with an asterisk (*) are also identified as interfacing residues. While

red indicates mutations that causes loss of binding to FI, light red indicates those with reduced binding to FI. Other residues selected for mutation but are not part of the complex interface are coloured purple and pink. Purple indicates mutations with reduced binding to FI, and pink highlights mutations with no significant change to the binding affinity with FI.

FI and 15 for PmSLP-1) and seven inter-protein peptides. When mapped onto appropriate protein models, all intra-protein crosslinks fall within the $C_a$-$C_a$ distance cut-off of 30 Å (S9B and S9C Fig) [33]. Upon mapping the inter-protein crosslinked lysine residues onto the cryo-EM model, we found that all the crosslinks localized to the proposed binding site, and the measured distance between most cross-linked lysine-lysine pairs was also within the 30 Å limit (Fig 4A).

In addition to the XL-MS data, we also identified several residues on PmSLP-1 that are important for its function. From the cryo-EM structure of PmSLP-1 in complex with FI, we mutated ten solvent-exposed residues on PmSLP-1 that reside at or near the complex interface and evaluated their roles in FI binding. We found that V214D and E240A completely lost the ability to bind bovine FI, while I133D, K236E, and E132R had no effect on protein binding. The remaining mutations reduced binding to FI but to varying degrees (Figs 4B and S10 and S6 Table). The distribution of the mutated residues on the surface of PmSLP-1 is depicted in Fig 4B (right panel). Altogether, these structural studies greatly complement each other, revealing the molecular details of the interaction between PmSLP-1 and bovine FI.

## PmSLP-1 promotes factor I degradation of C3b and C4b

Given that PmSLP-1 interacts with FI in a similar manner to the native co-factor FH, we next asked whether PmSLP-1 could also activate and promote FI-mediated degradation of C3b and C4b. To this end, we incubated purified human C3b and C4b with either PmSLP-1, bovine FI, or PmSLP-1:FI complex at 37°C for 30 min. C3b and C4b amounts were present at 12.5 times molar excess compared to FI and PmSLP-1. Neither FI nor PmSLP-1 alone was able to cleave C3b and C4b, indicating that FI and PmSLP-1 lack intrinsic proteolytic activity against these molecules under the tested conditions (Fig 5A and 5B). In contrast, when FI was pre-incubated with PmSLP-1, we observed characteristic cleavage profiles for both C3b (60 and 43 kDa fragments) and C4b (C4d and a 25 kDa fragment). This degradation pattern was also observed when FI was incubated with its endogenous co-factors, FH and C4BP (Fig 5A and 5B). These results demonstrate that PmSLP-1 possess a co-factor activity for FI. Moreover, while FH and C4BP have preference for specific substrates [34,35], PmSLP-1 could promote FI-mediated degradation of both C3b and C4b.

The serine protease active site irreversible inhibitors diisopropylfluorophosphate (DFP) and phenylmethylsulfonylfluoride (PMSF) are highly dependent on the catalytic site triad residues (Asp, His, Ser) having the correct hydrogen bonding alignment that allows the active site serine hydroxyl to nucleophilically attack these reagents. It has long been known that factor I, despite having the sequence characteristics of a serine protease, fails to be inactivated by incubation with DFP or PMSF [36]. However, when the preincubation of FI with DFP is done in the presence of the substrate C3b, DFP incorporation at the active serine of FI is achieved and this results in the inhibition of FI proteolytic activity when FH is subsequently added to the DFP-treated FI plus C3b mix. By contrast, preincubation of FI with DFP in the presence of FH does not result in the incorporation of DFP into FI [37]. Interestingly, upon treating the complex of FI bound to PmSLP-1 with PMSF, the complex completely loses its ability to cleave substrates (Fig 5C-E). However, when FI is present by itself, as expected, PMSF is unable to inhibit the enzymatic activities. This is seen by the rescue of cleavage activity following the addition of PmSLP-1 to the PMSF-treated FI sample. These results suggest that the high-affinity binding of PmSLP-1 to FI induces a conformational change in the catalytic domain of FI, which not only promotes FI-mediated degradation of C3b/C4b but also allows PMSF to access and irreversibly inactivate the serine protease. A closer look at the catalytic triad of FI, while bound to PmSLP-1, reveals that we were able to observe density at this site in the cryo-EM structure, which was missing in the crystal structure of FI alone (Fig 5F) [38]. Thus, this stabilization of the catalytic domain appears to be dependent on the cofactor (PmSLP-1) and does not require the presence of the substrate. This contrasts with the situation with the

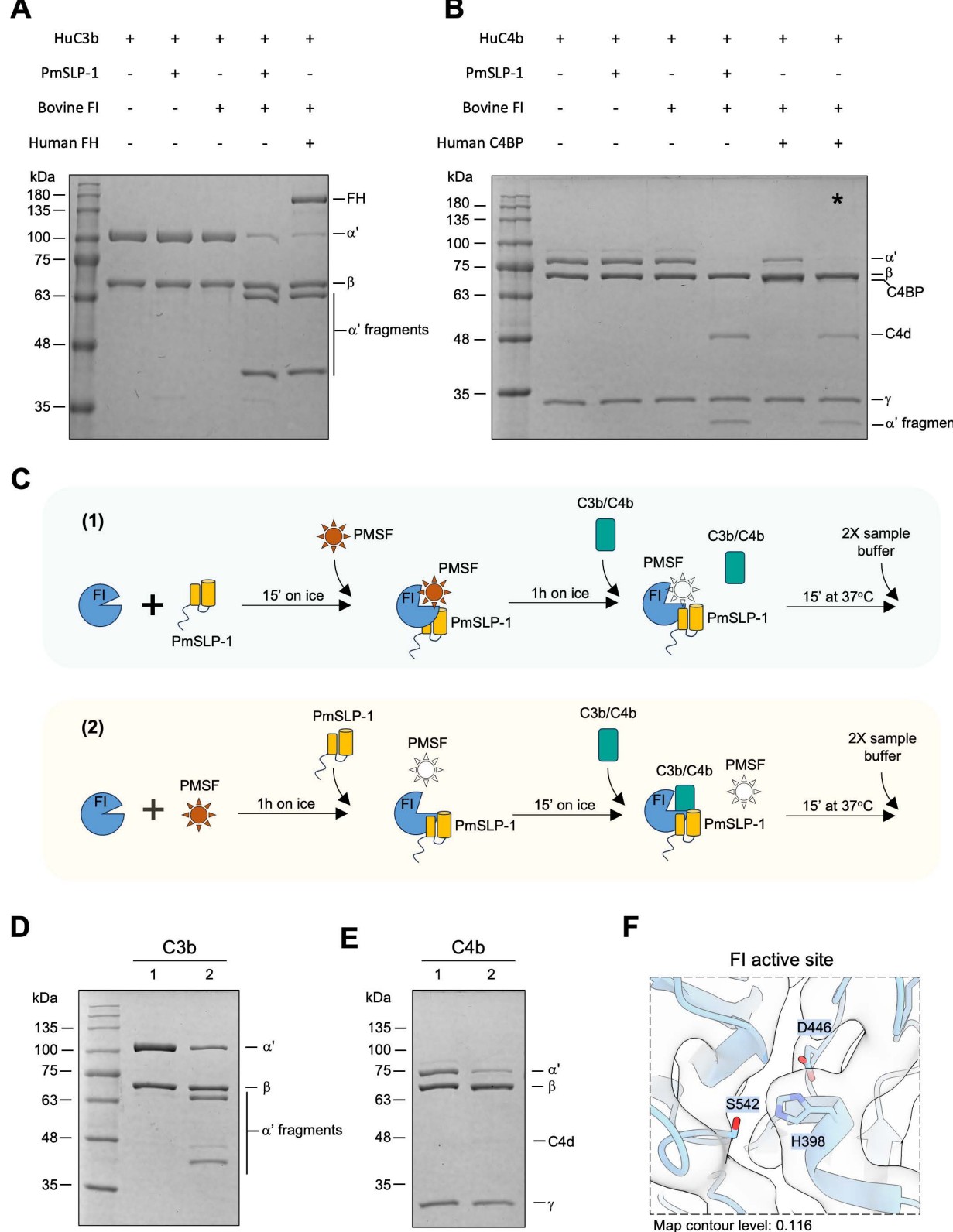

**Fig 5. PmSLP-1 activates factor I and promotes cleavage of C3b and C4b.** (A-B) FI-mediated degradation of C3b (A) and C4b (B) were evaluated in the presence of PmSLP-1 or the native fluid-phase cofactors (factor H and C4BP). The * in panel (B) indicates that human FI was used instead of

bovine FI as there appeared to be a species incompatibility between human C4BP and bovine FI that resulted in much diminished FI-mediated cleavage of C4b to C4c and C4d in the neighboring lane to the left. (C) Protocol used for testing the effects of PMSF on FI. PMSF is added to either the pre-formed PmSLP-1:FI complex (1) or FI alone (2). After 1h incubation with PMSF, PmSLP-1 is added to group (2) to allow for complex formation. The substrates, C3b or C4b, are then added to the mixture, and the reaction is stopped by the addition of 2X sample buffer. Active PMSF is depicted as a red star, whereas inactive (i.e., water-hydrolysed) PMSF is shown as a blank star. (D-E), FI degradation of C3b (D) and C4b (E) in the presence of PmSLP-1. Lane 1 and 2 in both panels correspond to the treatment described in panel (C). N = 3 biologically independent experiments. (F) Detailed view of the catalytic triad of bovine FI. The cryo-EM map is depicted as clear surface with black outline.

natural cofactor, FH, where its weak interaction with FI does not on its own induce the appropriate arrangement of the catalytic triad of FI, but a similarly weak binding of FI to its substrate C3b, in the absence of FH, does [37].

### Surface-exposed PmSLP-1 enhances serum resistance in *E. coli*

Thus far, we have demonstrated that PmSLP-1 could activate FI and promote cleavage of C3b and C4b *in vitro*. Next, we sought to evaluate the role of PmSLP-1 *in vivo* through a gain-of-function experiment. Specifically, we examined whether surface display of PmSLP-1 in a serum-sensitive *E. coli* strain could confer bacterial resistance to complement-mediated killing. *E. coli* C43 expressing proteins of interest were incubated with either 15% (v/v) normal bovine serum or heat-inactivated bovine serum. Optical density at 600 nm was used to evaluate bacterial growth. All cells exhibited similar growth in heat-inactivated serum conditions, but only cells expressing both Slam+PmSLP-1 were able to survive in normal bovine serum (Fig 6). These findings demonstrate that surface expression of PmSLP-1 alone was sufficient to provide protection against complement-mediated killing.

## Discussion

The complement system is an effective innate defence mechanism against invading microbes. This intricate network of proteins, when activated, rapidly removes target cells through phagocytosis and cell lysis. In response, bacterial pathogens have evolved numerous strategies to evade complement activities. Although researchers have identified numerous key virulence factors in *P. multocida*, the capsule is the only known factor that aids in serum resistance [9,10,24]. In this study, we have identified a novel mechanism used by *P. multocida* to prevent complement-mediated killing. We have shown that a subset of *P. multocida* serogroup A or BRD-causing strains are capable of binding to complement regulator

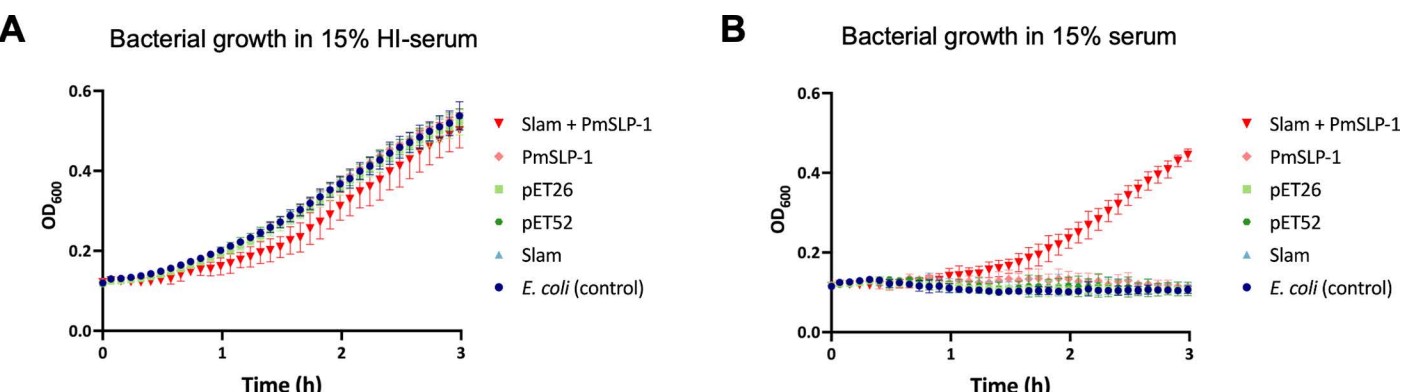

**Fig 6. Expression of PmSLP-1 on *E. coli* cell surface is sufficient to mediate serum resistance.** (A-B) *E. coli* cells expressing PmSLP-1, Slam, Slam+PmSLP-1, or empty vectors were grown in LB media supplemented with either heat-inactivated (A) or normal bovine serum (B). Bacterial growth at 37°C was monitored for 3 hours and OD$_{600}$ readings were recorded every 5 minutes. All values plotted represented the mean ± SD from three biologically independent experiments, each of which includes two technical replicates.

factor I, making this the first report of complement acquisition by *P. multocida.* Previously, we had named this lipoprotein PmSLP-1 to distinguish it from other PmSLP variants. However, considering our recent findings, we propose to refer to this protein as FI-binding protein (fIbp) to reflect the function of this protein.

FI is a key player in the inactivation of the C3 convertases in all three complement pathways. This serine protease cleaves the C3b and C4b subunits of these convertases into their inactive forms, a process that is normally facilitated by the presence of FI cofactors, such as FH and C4BP [39]. Due to its significance to complement regulation in the host, many microbial species have evolved with the ability to sequester and use FI to their own advantage. Recently, a glycoprotein ISG65 from *Trypanosoma brucei* was found to stimulate FI-mediated conversion of C3b to iC3b, thereby inhibiting the deposition of C3b onto *T. brucei* cell surface [40]. The authors suggested that the binding of ISG65 to C3b induced a conformational change in C3b that allows FI to cleave C3b more readily. However, the presence of co-factors, FH or CR1, is still required, as ISG65 alone could not promote the degradation of C3b by FI. Similarly, two Gram-negative pathogens, *Prevotella intermedia* and *Acinetobacter baumannii,* have also been shown to stimulate FI-mediated cleavage of C3b and C4b, but through a different mechanism [41,42]. Malm *et al.* demonstrated that *P. intermedia* could recruit not only FI, but also FH and C4BP, to its surface. However, the protein or proteins responsible for binding these complement factors remain unknown [41]. In contrast, in *A. baumannii*, a sole protein (CipA) was found to directly bind FI. The authors also proposed that CipA could form a quadripartite complex with FI, the cofactors, and the substrates (CipA:FI:C3b:FH or CipA:FI:C4b:C4BP) to facilitate the degradation of the substrates [42]. Acquisition of FI as an immune evasion strategy has also been discovered in a Gram-positive pathogen, *Staphylococcus aureus* [43,44]. Specifically, clumping factor A (ClfA) was shown to directly bind to FI and increase the cleavage of C3b to iC3b. Interestingly, FI-bound to ClfA was able to target C3b even in the absence of FH, suggesting that ClfA acts as a surrogate co-factor of FI. Our study shows that PmSLP-1 shares the same molecular mechanism with ClfA, whereby PmSLP-1 interacts directly with FI and stimulates the cleavage of C3b and C4b. While we did not investigate whether PmSLP-1 forms direct contact with the substrate, we suspect that the arrangement of PmSLP-1:FI:C3b would differ from that of FH:FI:C3b, based on our docked model of the complex. Additionally, since neither C3b nor C4b was observed in our pulldown experiments, the interaction between PmSLP-1 and substrates, if they exist, is likely transient.

Through *in vitro* binding assays, we demonstrated that PmSLP-1 formed a tight complex with FI, exhibiting a binding constant in the low nanomolar range. As there are several of the host's complement proteins that would compete for binding to FI, having the ability to sequester and interact tightly with FI would be a huge advantage for *P. multocida*. Additionally, FI is present at a much lower concentration in serum compared to other complement proteins [45]. Therefore, it would be beneficial for bacteria, like *P. multocida,* to sequester these molecules during an infection to maximize the protection against complement-mediated killing. Although we did not directly evaluate the effects of PmSLP-1 deletion on the pathogenicity of *P. multocida,* we showed through our gain-of-function experiments that the presence of PmSLP-1 alone on bacterial cell surface was sufficient to prevent complement-mediated killing. Thus, we would argue that PmSLP-1 is indeed a virulence factor in *P. multocida*.

Previous studies on the structure of FI in complex with FH and C3b highlight three key steps in the activation process: 1 - FH binds to C3b and induces a conformational change in C3b that exposes the scissile bond for cleavage, 2 – the CTC domain of C3b undergoes a marked rotation needed for FI binding, and 3 – FI forms contacts with both FH and C3b causing a rigid body rotation of the heavy chain and stabilization of the serine protease (SP) domain, both of which are required for FI to cleave C3b [31]. In our cryo-EM structure, we observed density for the catalytic triad of FI, which was not resolved in the crystal structure of human FI in its free form. It is likely that the binding of PmSLP-1 to FI helps to stabilize this highly dynamic SP domain. While we did not investigate the structure of PmSLP-1 in complex with FI and the substrate C3b, it would not be surprising to observe C3b presenting a different conformation compared to its structure in complex with FH and FI. Our structural data also revealed that PmSLP-1 shares the same binding interface on FI with FH, but the high affinity binding between PmSLP-1 and FI in the absence of the substrate highlights a key difference in the

mode of action between PmSLP-1 and FH. The fast on-rate and slow off-rate for the interaction between PmSLP-1 and FI also suggests that PmSL-1 would be quite competitive against FH in acquiring host FI during a bacterial infection. Overall, these results highlight the importance of PmSLP-1 to *P. multocida* pathogenesis and provide structural insights that would greatly benefit the development and engineering process of PmSLP-1 as a vaccine antigen. In addition, since PmSLP-1 could bind to FI from other ruminants and *P. multocida* has been reported to cause infection in both sheep and goat [46], it is possible that a PmSLP-1 based vaccine could also protect these animals from *P. multocida* infection.

Based on our data, we proposed a model to explain the molecular mechanism of complement inactivation by PmSLP-1 (Fig 7). A subset of *P. multocida* strains expresses PmSLP-1 on their outer membrane surface. PmSLP-1 not only exhibits tight binding to the host's FI, but it is also capable of stabilizing the catalytic domain of FI. This interaction allows FI to remain in its active form, primed and ready to cleave nearby C3b and C4b molecules. Consequently, the degradation of C3b and C4b would effectively prevent further deposition of C3b by all three complement pathways. Thus, by blocking the complement cascade earlier on, the bacteria would more likely be able to survive and propagate. This perhaps explains the severity of *P. multocida*-associated infection and diseases in animals.

The first step to survival within the harsh environment of a mammalian host for invading bacteria is to escape the complement system unscathed. Consequently, it is not uncommon to observe bacterial pathogens with more than one defense strategy against complement-mediated killing. *N. meningitidis* and *S. aureus* are classic examples of such pathogens [11,47]. Both pathogens are capable of binding to multiple complement regulators, and the recruitment of complement inhibitors are often carried out by bacterial surface proteins. Therefore, it is possible that binding of FI is not the sole mechanism employed by *P. multocida*, and further studies are needed to discover other immune evasion strategies in this zoonotic pathogen.

## Methods

### Plasmids, antibodies, and protein sequences

See S7 Table for a summary of plasmids and antibodies used in this study.
See S8 Table for the amino acid sequences of the main proteins of this study.

### Data analysis and figure preparation

Figures were created using UCSF ChimeraX v.1.0.0 (ref. [48]). Data from biolayer interferometry experiment, thermal shift assay, and bacterial growth assay were analyzed and visualized with Prism v.10.0.2 on Mac (GraphPad Software, www.graphpad.com).

**Expression and purification of PmSLP-1.** *pmSLP-1* gene was PCR-amplified from genomic DNA of *P. multocida* strain 36950, and PmSLP-1 construct was cloned into a pET-52b vector encoding a thrombin cleavage site and a 6x-His tag downstream of the insertion site of *pmSLP-1*. To promote stable cytoplasmic expression of the PmSLP-1 in *E. coli*, the N-terminal signal peptide and the first 15 residues of PmSLP-1 were removed. Plasmids carrying PmSLP-1[15-318] constructs were transformed into competent *E. coli* strain T7 SHuffle (New England Biolabs), and transformants were selected on Luria-Bertani (LB) agar with 100 μg/mL ampicillin. Multiple colonies were used to inoculate 20 mL of LB with 100 μg/mL ampicillin and grown at 37°C for 16 hours. The overnight cultures were subsequently used to inoculate 2 L of LB supplemented with 100 μg/mL ampicillin. Cell cultures were grown shaking at 175 rpm at 37°C to mid-log phase, and protein expression was induced with isopropyl-D-thiogalactosidase (IPTG, 0.5 mM final concentration). Cells continued to grow overnight at 20°C. Cells were pelleted at 6,000 $x g$ and resuspended in 40 mL of lysis buffer (50 mM Tris-HCl pH 8.0, 300 mM NaCl) with 10 mM imidazole, supplemented with 1 mM phenylmethylsulfonyl fluoride, 1 mM benzamidine, 1 mg/mL lysozyme, and 0.03 mg/mL DNase I. Cells were lysed by sonication for 10 mins and centrifuged at 35,000 $x g$ to remove cell debris. The supernatant was filtered through a 0.45 μm filter and incubated at 4°C for 16 hours with 2 mL HisPur

## Alternative Pathway

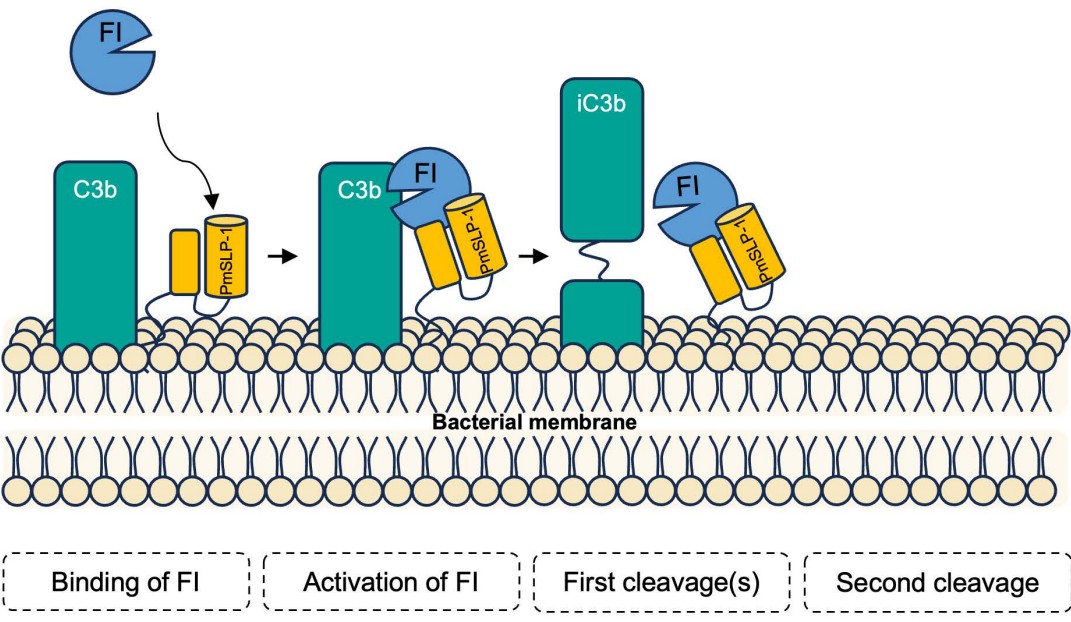

## Classical pathway & Lectin pathway

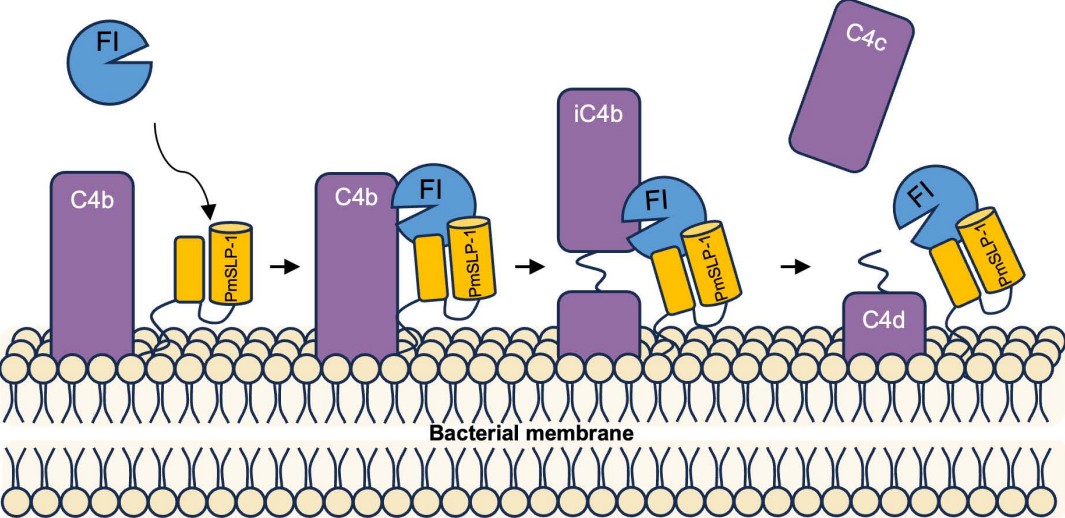

**Fig 7. Proposed mechanism of PmSLP-1-mediated immune evasion in *P. multocida*.** PmSLP-1 present on the surface of *P. multocida* hijacks complement factor I from the host. The binding of PmSLP-1 to FI induces a conformational change, locking FI in the active state. The PmSLP-1-bound FI cleaves C3b and C4b into their inactive forms in a similar manner observed when factor H or C4BP is present. In the presence of PmSLP-1, the FI-mediated processing of C3b halts after the production of iC3b, whereas the cleavage of C4b proceeds to the C4c and C4d formation. In both cases, the effective clearance of C3b and C4b molecules allows the bacteria to avoid killing from all three complement pathways.

Ni-NTA resin (Thermo Fisher Scientific). Beads were pelleted for 5 min at 700 $x\,g$, loaded onto a gravity column (Econo-Pac Bio Rad), and washed with 100 mL of cold wash buffer (lysis buffer with 20 mM imidazole). Protein was eluted in 10 mL of cold elution buffer (lysis buffer with 400 mM imidazole) and dialyzed for at least 16 hours at 4°C against 1 L of 20 mM Tris-HCl pH 8.0 and 100 mM NaCl. When removal of the carboxyl terminal His-tag was needed, the protein was incubated with 2 units of bovine thrombin (Sigma Aldrich) in the same dialysis buffer as above. Then, 100 µL of HisPur Ni-NTA resin (Thermo Fisher Scientific) and 100 µL of $p$-aminobenzamidine-agarose (Sigma Aldrich) were added to the sample to remove the His-tag and thrombin. Protein sample was concentrated with a 10K MWCO concentrator (Sartorius) and further purified by size exclusion chromatography on a Superdex 200 Increase 10/300 GL column, equilibrated in PBS (Cytiva).

**Crystallization, diffraction data collection, and structure building of PmSLP-1.** The first 94 residues following the N-terminal signal peptide were predicted to be highly disordered [23]. Thus, to aid in crystallization of PmSLP-1, these residues along with the cognate signal peptide were removed. Additionally, mutations of three charged residues E315A, K316A, and K317A, predicted by the UCLA MBI SERp server, were made. Plasmids containing the PmSLP-1[95-318] SERp were transformed into *E. coli* T7 Express (New England Biolabs) or m384 for selenomethionine protein. Expression and purification of PmSLP-1[95-318] SERp in *E. coli* T7 Express followed the protocol described above. M384 transformants were selected on LB agar with 100 µg/mL ampicillin, and colonies were used to inoculate 50 mL of 0.02 mg/mL methionine supplemented minimal media (M9 with final 0.2% glucose, 1 mM $MgSO_4$, 1 mM thiamin, 0.02 mg/mL essential L-amino acids) with 100 µg/mL ampicillin. Cells were grown at 37°C with shaking for 16 hours and used to inoculate 2L of minimal media supplemented with 0.02 mg/mL selenomethionine. Large cell cultures were grown at 37°C with shaking until mid-log phase. Protein expression was then induced with addition of IPTG to a final concentration of 0.5 mM, and cells continued to grow at 20°C overnight. After cells were collected and lysed, Se-Met PmSLP-1[95-318] SERp protein was purified using the same protein purification protocol. Purified PmSLP-1[95-318] SERp and Se-Met PmSLP-1[95-318] SERp proteins at 20 mg/mL were screened for crystallization at 22°C with 1:1 (protein:precipitant) ratio. Protein crystals were observed in 0.2 M ammonium tartrate with 30% PEG 8000 and in 0.1 M sodium thiocyanate pH 6.5, 0.2 M ammonium sulfate, with 20% PEG 3350. Subsequent streak seeding helped to increase crystal sizes and decreased crystallization time of Se-Met PmSLP-1[95-318] SERp. To prepare protein crystals for data collection, crystals were collected and cryo-protected in reservoir condition supplemented with 20% glycerol for 30 sec or soaked in 1 µL of cryo-protectant with 0.5 M NaBr or with resuspended selenourea crystals for 5 mins [49].

Native and anomalous data were collected on vitrified crystals at 105K on 08ID-1 beamline at Canadian Macromolecular Crystallography Facility at the Canadian Light Source. A 900-image dataset was collected on all crystals at the anomalous peak wavelength for selenourea-soaked PmSLP-1[95-318] crystals as determined by a fluorescence scan. Data processing of a selenourea-soaked SeMet crystal by XDS [50] revealed anomalous measurability to 2.9 Å resolution, based on a SigAnom value > 1. PHENIX AutoSol [51] was used for phasing, density modification, and to build an initial model. The phasing solution has a figure of merit of 0.226, an overall score of 42 ± 11, and 7 selenium sites identified. For the refinement, the dataset was cut at 2 Å using a $CC_{1/2}$ cutoff at 50% or higher. The final model was generated following several rounds of model building and refinement using COOT [52] and PHENIX refine [53], resulting in a final $R_{work}/R_{free}$ of 0.203/0.244. Data and model statistics are shown in S1 Table, generated with Phenix.table_one.

**Bovine FI purification.** Bovine complement FI was purified from Gibco bovine sera (Thermo Fisher Scientific). The purification method was adapted from Menger and Aston [54]. In brief, sera (250 mL per purification) were centrifuged at 3500 x $g$ to remove insoluble components and dialyzed in 4 L of 0.02 sodium phosphate pH 8.0 containing 0.01 M EDTA and 0.12 M NaCl (Buffer A) overnight at 4°C. Dialyzed sera were spun down at 5000 x $g$ for 10 mins to remove precipitates and filtered through a 0.22 µm filter. Clarified sample was subsequently loaded onto a Q-Sepharose column equilibrated with Buffer A. Unbound fractions were collected and fractionated with $(NH_4)_2SO_4$. The precipitate formed between 40–60% saturation of $(NH_4)_2SO_4$ was dissolved in and dialyzed against 0.02 M potassium phosphate pH 6.0 with

0.02 M NaCl (Buffer B) overnight at 4°C. Dialyzed sample was loaded onto a HiTrap Capto S column (Cytiva) equilibrated with buffer B, and bound proteins were eluted by a linear gradient of 0.02 to 0.2 M NaCl, followed by 2 M NaCl. Fractions containing FI, determined by Western blotting with anti-human FI antibodies, were pooled and concentrated with a 30K MWCO concentrator (Sartorius). Concentrated sample was buffer exchanged into phosphate-buffered saline (PBS) pH 7.4 and incubated with purified PmSLP-1 conjugated to CNBr-Activated Sepharose 4B resin (GE Life Sciences) for 16 hours at 4°C. PmSLP-1-conjugated Sepharose resin was packed in a gravity column (Econo-Pac Bio Rad) and washed with at least 50 mL of PBS. Complement FI was eluted off the column with 5 mL of 0.1 M glycine buffer pH 2.2 and immediately neutralized with 1 M Tris pH 9.0 (10% v/v). Purified protein was dialyzed against PBS for subsequent assays.

**Structural determination of PmSLP-1:FI complex using cryo-EM single particle.** Purified PmSLP-1 and FI were mixed to achieve a final concentration ratio of 5:1 and incubated on ice for 10 mins before loading onto a Superdex Increase 200 10/300 GL column (Cytiva). The column was pre-equilibrated in 1X PBS pH 7.4. Fractions containing the protein complex were determined with SDS-PAGE analysis and concentrated to 0.25 mg/mL. Then, 3.5 µL of sample was applied to a glow discharged UltrAufoil holey gold grids (R2/2, 200 mesh, Quantifoil). Grid were blotted for 5 s at 4°C and 95% relative humidity before freezing in liquid ethane using a Vitrobot Mark IV (FEI). The cryo-EM dataset was collected at the Pacific Northwest Cryo-EM Center using the Serial-EM automated data acquisition software on a Titan Krios (FEI) operated at 300 kEV and equipped with a Falcon III direct electron detector (Thermo Fisher Scientific) and a BioQuantum K3 imaging filter (Gatan). A total of 2,335 movies was acquired at a 40° tilt in super-resolution mode with a pixel size of 0.411 Å, a total dose of 50 e⁻, and a defocus range of -0.8 to -2.0 µm.

**Cryo-EM image processing.** All image analysis was performed with cryoSPARC v3 (Ref [55]), and the data-processing procedure is outlined in S5 Fig. In brief, movies were aligned with patch-based motion correction, followed by patch-based CTF estimation. The dataset was manually curated by removing movies with poor CTF fit (> 6 Å) and visible signs of devitrification and/or large ice contamination, leaving 1,948 images for further analysis. Particles were picked by using blob-picker with the minimum and maximum particle diameters of 75 Å and 300 Å, respectively. An initial 1,938,372 particles were picked and extracted with a box size of 512 pixels and Fourier-binned to 128 pixels. Several rounds of 2D classification, followed by *ab initio* and heterogenous refinement were done to remove 'junk' particles. Rebalance 2D classes job was used to identify projections with low distribution, and Topaz was used to search for more particles belonging to these classes [56,57]. After combining the particle stacks and removing duplicates, a round of heterogenous refinement was used for further cleanup. The final particle stack containing 687,259 particles was used in homogenous refinement, resulting in a 3.9 Å map. The map indicates that the protein complex exists as a dimer. To further improve the resolution, particles coming from micrographs with a CTF fit > 4 Å were removed from the particle stack. The remaining 343,391 particles were re-extracted with a box size of 540 pixels and binned to 180 pixels. Another round of heterogenous refinement revealed two conformations of the dimer with a slight variation in the relative orientation of the protomers. Both particle stacks were subjected to non-uniform refinement ($C_2$), symmetry expansion ($C_2$), particle subtraction, and local refinement, which resulted in two particle stacks of the protomer, i.e., the monomeric complex. These particles were combined and further processed with local refinement and 3-D classification. A final stack of 200,426 particles was used in local refinement yielding a 3.5 Å $C_1$ symmetry map that was used for model building.

### Cryo-EM model building

ChimeraX [48] was used for initial manual docking of the AlphaFold2-predicted structure of PmSLP-1:FI complex into the EM map. The protein sequence for bovine factor I was obtained from Uniprot (A0A3Q1MF14_BOVIN). Several rounds of model building and refinement was done in COOT [52] and PHENIX real-space refinement with secondary structure and reference structure restraints enabled [53]. Data and model statistics are generated with PHENIX.validation_cryoem shown in S4 Table.

**Cross-linking mass spectrometry.** Purified PmSLP-1:FI complex prepared in PBS pH 7.4 was cross-linked with disuccinimidyl suberate (DSS, Thermo Fisher Scientific). Briefly, 4 µL of 25 mM DSS freshly prepared in DMSO was added to 100 µL of PmSLP-1:FI complex at 1 mg/mL. The crosslinking reaction was incubated at room temperature for 20 mins. To stop the reaction, 50 µL of 1M ammonium carbonate was added, and the mixture was incubated at room temperature for 30 mins. The sample was concentrated to 20 µL and mixed with equal volume of SDS loading buffer. Thirty µL of the final sample was analyzed on an 8% SDS-PAGE gel, stained with Coomassie Brilliant Blue. Protein band at roughly 120 kDa was excised and submitted to the Southern Alberta Mass Spectrometry Facility (University of Calgary, AB, Canada) for LC-MS/MS analysis. Protein band was digested with trypsin, and the peptides were analyzed on an EasyLC1000 nano-chromatography system coupled with an Orbitrap Velos mass spectrometer.

Data were analyzed using the Mass Spec Studio v2.4.0.3545. DSS cross-link residue pairs were constrained to only lysine (K) on both peptides. Higher energy collisional dissociation (HCD) fragmentation was used for MS acquisition. Default processing parameters were used: Trypsin digestion (K/R only), fragment charge states 1 and 2, peptide length 5–60, % E-value threshold = 99, MS mass tolerance = 10 ppm, MS/MS mass tolerance = 10, and elution width = 0.3 min. Peptide spectra matches were manually inspected for data quality and correct assignments of cross-linked residues. Finally, peptide pairs and residue pairs that satisfied a false discovery rate of 0.05 were exported and used for further structural analysis.

**Co-immunoprecipitation and mass spectrometry.** Modification was made on the previously described PmSLP-1 construct to replace the His-tag with a FLAG-tag, and the plasmid was transformed into *E. coli* T7 SHuffle. Transformants were selected on LB agar with 100 µg/mL ampicillin. Multiple colonies were used to inoculate 5mL of LB with 100 µg/mL ampicillin and grown at 37°C for 16 hours. The starter cultures were subsequently used to inoculate 50 mL of LB supplemented with 100 µg/mL ampicillin. Cell cultures were grown at 37°C to mid-log phase, and protein expression was induced with isopropyl-D-thiogalactosidase (IPTG, 0.5 mM final concentration). After 16-hour growth at 20°C, cells were pelleted at 3500 *x g* and resuspended in 1 mL of lysis buffer (50 mM Tris-HCl pH 8.0, 300 mM NaCl) with 10 mM imidazole. Cells were lysed by sonication for 10 mins and centrifuged at 10,000 *x g* to remove cell debris. The supernatant was incubated with 25 µL of Pierce anti-FLAG affinity resin (Thermo Fisher Scientific) at 4°C for 1 hour. The beads were pelleted for 5 min at 700 *x g* in 1.5 mL tubes. The supernatant was discarded, and the resin was washed 3 times with PBS pH 7.4. To ensure that PmSLP-1 remained bound to anti-FLAG resin, an aliquot of the resin was mixed with SDS loading buffer and analyzed on an SDS-PAGE gel. PmSLP-1-bound resin was incubated with 1 mL of 10% v/v normal bovine sera (diluted in PBS; Gibco Thermo Scientific) for 3 hours at 4°C. The resin was pelleted for 5 min at 700 *x g*, the supernatant was discarded. The resin was washed five times with PBS pH 7.4, and excess buffer was completely removed before freezing down the beads. Samples were submitted to the SPARC BioCentre (The Hospital for Sick Children, ON, Canada) for further analysis.

Protein samples were first eluted off the resin using FLAG peptide (GenScript) and digested with trypsin. Eluted peptides were then analyzed using liquid chromatography-tandem mass spectrometry (LC-MS/MS) on an EASY-nLC 1200 nano-LC system, coupled with an Orbitrap Fusion Lumos Tribid mass spectrometer (Thermo Scientific). MS/MS results were analyzed with Proteome Discoverer (MS Amanda 2.0 and Sequest HT) and Scaffold (X! Tandem), and the peptides were searched against the *Bos taurus* (UP000009136) and *Escherichia coli* (UP000000625) proteomic databases. Scaffold (v5.3.0) was used to analyze the MS/MS results. A probability of at least 95% was required for accepted peptide and protein identification, and a positive protein hit must also contain at least 2 identified peptides. The protein with the highest number of exclusive spectrum count and was exclusively found in the target protein, PmSLP-1, was considered a potential binding partner.

**Biolayer interferometry binding assays.** Purified bovine FI was biotinylated with the EZ-link NHS-biotin reagent according to the protocol provided by Thermo Fisher Scientific. Excess biotin was removed with a 10K MWCO concentrator (Sartorius). Binding experiments were performed using the Octet RED96 system (FortéBio). Biotinylated FI was

immobilized onto Streptavidin (SA) biosensors (Satorius) and used to measure binding activity to varying concentrations of analyte (wildtype and mutant PmSLP-1). All experiments were performed at 30°C at a shaking speed of 1000 rpm. SA biosensors were hydrated for 10 mins in kinetics buffer (PBS pH 7.4 and 0.01% Tween 20) and loaded for 60s with 150 nM biotinylated FI. Sensors were then washed in a kinetics buffer for 200 sec to establish a baseline. Various concentrations of analytes were used for the binding assay: 0, 5, 16, 50, 160, 500, and 1600 nM. For each concentration, the binding steps included 150 sec of association, 150 sec of dissociation, and 30 sec of regeneration. To regenerate, sensors were dipped into a buffer containing 100 mM sodium citrate pH 4.5 and 100 mM NaCl; this step allowed the removal of any analyte still bound to FI after the dissociation step. To control for non-specific interaction between the SA biosensors and the analytes, reference sensors without ligand loaded were subjected to the same binding steps with various concentrations of analyte as described above. Steady-state analysis was used to determine the binding affinity between the ligand and analyte. Saturation binding curves were derived by taking an average of the response values in the last 5 seconds of the association step and plotting these against the analyte concentration. Saturation curves were fitted with Prism (GraphPad) using the "One site – Total binding" saturation model assuming specific binding to a single site.

**Thermal shift assay.** The thermal shift assay was conducted using a Tycho NT.6 (NanoTemper) system. Purified protein samples were prepared at 1 mg/mL in 20 mM Tris pH 8.0 and 100 mM NaCl. The samples were subjected to a temperature ramp from 35°C to 95°C, and changes in intrinsic fluorescence intensities at 350 nm and 330 nm were recorded. The temperature inflection points ($T_i$) were obtained from the first derivative graphs reported by the program.

**E. *coli* surface lipoprotein translocation assay.** *E. coli* strain C43(DE3) cells were transformed with each plasmid listed in S7 Table. Cells were grown overnight at 37°C in LB media supplemented with either kanamycin or ampicillin. Overnight cell cultures were diluted with LB media, supplemented with appropriate antibiotics, to $OD_{600} = 0.5$, and 1 mL of each diluted culture was harvested by centrifugation at 5000 x *g*. Cell pellets were washed twice and resuspended in 200 µL of PBS. One hundred µL of resuspended cells were incubated with 100 µL of 1 mg/ml proteinase K (Sigma) while the remaining the cells were treated with PBS (negative control). After 30 mins of incubation at room temperature, cells were centrifuged at 3500 *x g* for 5 mins, and the supernatant were discarded. Cells were washed five times with PBS to remove any residual proteinase K and finally resuspended in 50 µL of PBS supplemented with 1 mM PMSF. Samples were mixed with equal volume of 2X SDS loading dye, boiled for 10 mins at 95°C, and analyzed via Western blot. Presence of intact PmSLP-1 was detected with the primary anti-FLAG antibodies and secondary anti-rabbit HRP antibodies (Thermo Fisher Scientific). Ponceaus S (Bioshop) was used to stain the nitrocellulose membrane to evaluate the total amount of protein loaded per lane.

**Bacterial growth assay.** *E. coli* strain C43(DE3) cells transformed with each vector listed in S7 Table were grown as described in the *E. coli* surface lipoprotein translocation assay. Twenty-five µL of cells at $OD_{600} = 0.5$ were incubated with 75 µL of 15% normal bovine serum (v/v, diluted with LB). Bovine plasma was collected from healthy cattle at the University of Calgary (AB, Canada) and kept at -80°C until use. To inactivate complement proteins, aliquots of bovine serum were heated at 56°C for 30 mins and centrifuged at 21000 *x g* for 10 mins to remove precipitates. The heat inactivated serum was used as a negative control. Bacterial growth at 37°C in the presence of normal or heat-inactivated bovine serum was monitored in a 96-well plate using the Biotek Cytation5 plate reader. Optical density measurements at 600 nm were recorded every 5 mins for 3 hours. The assay was repeated with 3 biological replicates, each containing 2 technical replicates. The reported growth curves represent the average values of all replicates.

**Substrate degradation assay with bovine FI.** fPmSLP-1 and bovine FI were purified as described above. While the use of bovine complement proteins in this assay would be more meaningful, we proceeded with the human complement factors, including C1s, C3b, C4, FH, and C4BP, due to difficulties in purifying these proteins from bovine sera. Purified human complement proteins were either purified from human serum or purchased from Complement Technology, Inc. (TX, United States). C4b was made by incubating C1s with C4 (1:100 v/v) for 30 mins at 37°C. To assess the cleavage of C3b and C4b by bovine FI, 1 µL of C3b (5.7 µM) or C4b (5.3 µM) was incubated with: 1 µL of FI (0.4 µM), 1 µL of PmSLP-1

(0.4 µM), or 1 µL of FI (0.4 µM) + 1 µL of PmSLP-1 (0.4 µM). Treatment of C3b and C4b with bovine FI preincubated with either 0.5 µL of human FH (6 µM) or 0.5 µL of C4BP (6 µM), respectively, were used as positive control. Each reaction was topped up with PBS to reach a total volume of 6 µL and incubated for 15 mins at 37°C. Where needed, PMSF was added at a final concentration of 2 mM. The reactions were then terminated by the addition of 2X SDS loading buffer containing β-mercaptoethanol. Cleavage products of C3b and C4b were resolved on a 10% SDS-PAGE gel.

**Sequence alignments of FI homologues.** In the S6 Fig, the factor I sequences used in the alignment include *H. sapiens* FI (P05156), *M. musculus* FI (Q61129), *G. gallus* FI (A0A8V0XKE8), *S. scrofa* FI (A0A287AQ20), *B. taurus* FI (A0A3Q1MF14), *O. aries* FI (W5P5I3), and *C. hircus* FI (A0A452DTU9). The sequences were obtained from Uniprot [58] with the accession codes listed above, aligned in Clustal Omega [59], and visualized in JalView 2.11.3.2 (ref. [60]). The phylogenetic tree was calculated using BLOSUM62 and constructed in JalView. The pairwise alignment score for each pair of protein sequence was calculated in JalView, and the scores were used to generate the percent identity matrix in Microsoft Excel.

## Supporting information

**S1 Table. Crystallographic data collection and refinement statistics for PmSLP-1.**
(TIF)

**S2 Table. Interactions between the handle domain and the barrel domain of PmSLP-1.** The analysis was done with the PISA analysis software.
(TIF)

**S3 Table. Representative LC-MS/MS results for the co-immunoprecipitation assay.** Transferrin binding protein B (TbpB) was used as a negative control. The table includes the name of the identified proteins, the molecular weight, and the spectral counts for each identified protein in the target PmSLP-1 sample and in the negative control sample. The hits were sorted in a descending order based on the spectral count in PmSLP-1 sample. Only the top 10 out of 567 hits were shown. The protein with the highest spectral count in the PmSLP-1 sample was highlighted.
(TIF)

**S4 Table. Cryo-EM data collection, refinement, and validation statistics for PmSLP-1:FI complex.**
(TIF)

**S5 Table. Interactions between bovine FI and PmSLP-1.** The analysis was done with the PISA analysis software.
(TIF)

**S6 Table. Kinetic and steady state parameters measured by biolayer interferometry.** Data were analyzed with the Octet Data Analysis software 7.0. For each interaction, a kinetic $K_D$ and a steady state $K_D$ were obtained. NB indicates no binding.
(TIF)

**S7 Table. List of plasmids and antibodies in this study.**
(TIF)

**S8 Table. List of amino acid sequences of proteins in this study.** The accession code for each protein is provided. The underlined sequence indicates the signal peptide.
(TIF)

**S1 Fig. Structural features of Slam-dependent SLPs.** (A-C) High resolution structure of representative Slam-dependent surface lipoproteins: PmSLP-1 from *Pasteurella multocida* (yellow), hemoglobin receptor from *Kingella dentrificans* (purple),

and factor H binding protein from *Neisseria meningitidis* (orange). The colour scheme is maintained throughout. The N- and C-termini of each structure are denoted with 'N' and 'C', respectively. The proteins share similar structural composition, including an 8-stranded β-barrel domain and an N-terminal handle domain made up of 5–6 β strands. Structural similarity between HpuA or fHbp and PmSLP-1 were evaluated in ChimeraX (the alignment is based on secondary structure scoring only) and the reported RMSD values for each domain are indicated above. Structural alignment of the handle domains is shown in (D) and (E). Structural alignment for the barrel domains is shown in (F) and (G).
(TIF)

**S2 Fig. Interaction between PmSLP-1 and FI is host specific. (A)** Phylogenetic tree and **(B)** percent identity matrix of FI homologues from various species. Sequence alignment was performed with ClustalW, and the average distance between each tree node was computed with BLOSUM62.
(TIF)

**S3 Fig. Characterizing the interaction between PmSLP-1 and bovine FI. (A)** An overlay of SEC elution profiles of purified PmSLP-1 (orange), bovine FI (purple), and a mixture of purified PmSLP-1 and bovine FI (5:1 molar ratio, black). The $V_0$ indicates the void volume. **(B)** SDS-PAGE analysis of the fractions from peak 1 and 3 of the SEC analysis of PmSLP-1 and bovine FI mixture. **(C-D),** Thermal stability profiles of PmSLP-1$^{15}$ and PmSLP-1$^{95}$ analyzed by nanoDSF. Changes in the protein intrinsic fluorescence signals under thermal stress (from 35$^{\circ}$C to 95$^{\circ}$C) are plotted with the raw $F_{350nm}$:$F_{330nm}$ ratio (C) and with the first derivative of the $F_{350nm}$:$F_{330nm}$ ratio (D).
(TIF)

**S4 Fig. Structural studies of PmSLP-1:FI complex. (A)** Purified PmSLP-1 and bovine FI were mixed at a 5:1 molar ratio and subjected to SEC analysis. **(B)** Fractions collected from peak 1 were analyzed on a non-reducing SDS-PAGE gel. Samples from these fractions were pooled, concentrated, and used to prepare Cryo-EM grids. "In1" denotes purified bovine factor I alone, and "In2" contains the input mixture of purified bovine FI and PmSLP-1 prior to gel filtration analysis. **(C)** AlphaFold2 model of PmSLP-1:FI, coloured by the pLDDT score. The full-length protein sequences (excluding the signal peptide sequence) of the mature proteins were used as inputs in AlphaFold2 (right panel). The N-terminus of both proteins were predicted to be disordered and were removed from the model during the model building process. **(D)** Cryo-EM 3D classes of the PmSLP-1:FI sample showing a two-fold symmetry indicating dimerization of the protein complex. The AlphaFold2 predicted model were docked into the cryo-EM maps to demonstrate that the PmSLP-1:FI complex dimerization occurs through FI. Overlaying the docked models and aligning them on protomer 1 also revealed a 17$^{\circ}$ rotation of protomer 2 in dimer 2 (grey) relative to protomer 2 in dimer 1 (teal).
(TIF)

**S5 Fig. Cryo-EM single-particle analysis of PmSLP-1 in complex with bovine FI. (A)** A visual representation of the image processing workflow used to reconstruct the structure of PmSLP-1:FI complex. CryoSPARC was used to obtain the final consensus, local refined maps. **(B)** The final reconstructed model of PmSLP-1 (yellow) in complex with bovine FI (blue) shows minor differences to the AlphaFold2 predicted model (pink). The model-to-map FSC curve (middle) calculated in Phenix and the 3D FSC plot (right) show how well the model fits in the cryo-EM map and the quality of the consensus map, respectively.
(TIF)

**S6 Fig. Multiple sequence alignment of factor I homologues.** The alignment was performed with ClustalW and colored according to the convention. Bovine FI (bolded) was used as the reference sequence. The arrows highlight the residues in bovine FI that form salt bridges with PmSLP-1.
(TIF)

**S7 Fig. Structural comparison between PmSLP-1:FI complex and FH:FI complex. (A)** Overlay structures of human mini-FH:FI complex (adapted from PDB: 5O32) and PmSLP-1:FI complex showing that FH and PmSLP-1 share the same binding interface on FI. Protein interfaces were analyzed with the PISA software (as part of the CCP4 package), and the interface parameters for each complex were reported. **(B)** Crystal structure of human mini-FH:FI:C3b (left, PDB: 5O32) was used to guide the modeling of PmSLP-1:FI:C3b (right). Surface representation of C3b (white) was used for clarity. (TIF)

**S8 Fig. Modeling the binding of PmSLP-1:FI complex to C3b. (A)** Ternary complex of PmSLP-1:FI:C3b modelled based on the crystal structure of human mini-FH:FI:C3b. Inset shows a detailed view of the clashes between PmSLP-1 and the MG6 domain of C3b (left panel) and the catalytic triad of bovine FI and the first scissile bond on C3b (right panel). **(B)** AlphaFold3 model of PmSLP-1:FI:C3b. The left inset shows the interface between PmSLP-1 and the MG6 domain of C3b. The right inset shows the AF3 structure of PmSLP-1 alone highlighting the incorrectly predicted handle domain, coloured in purple. **(C)** A representative cryo-EM map of the PmSLP-1:FI complex dimerizing through the heavy chain of FI (i). Two copies of the solved cryo-EM structure of PmSLP-1:FI monomer were docked into the cryo-EM dimer map, and the docked model is represented as a coloured surface (ii). The AF3 predicted model for the PmSLP-1:FI:C3b complex was combined with the solved cryo-EM structure of PmSLP-1:FI to create a new ternary model. PmSLP-1 and FI are shown as surface representation in yellow and blue, respectively; C3b is shown in cartoon representation (iii). The final panel (iv) shows an overlay of the PmSLP-1:FI dimer (ii) and the ternary complex model (iii) illustrating the clash between C3b on one protomer with FI of the other protomer. (TIF)

**S9 Fig. Cross-linking mass spectrometry analysis of the PmSLP-1:FI complex. (A)** SDS-PAGE analysis of the cross-linked sample. White box indicates the gel band that was excised and subjected to mass spectrometry analysis. **(B)** Intra-protein crosslinks mapped onto the crystal structure of PmSLP-1. The crosslinked residues, the number of occurrences per crosslink, and the estimated $C_a$ distance are reported. Crosslinked peptides containing residues in the disordered anchoring peptide of PmSLP-1 were excluded. **(C)** Intra-protein crosslinks mapped onto the solved cryo-EM structure of bovine FI. The crosslinked residues, the number of occurrences per crosslink, and the estimated $C_a$-$C_a$ distance are reported. (TIF)

**S10 Fig. Evaluating the binding affinity between bovine FI and PmSLP-1 mutants. (A-J)** BLI sensorgrams and saturation curves showing the binding between bovine factor I and PmSLP-1 mutants. Biotinylated bovine FI was immobilized onto streptavidin sensors, and binding was measured at various concentrations of PmSLP-1. The raw sensorgrams are representative of one replicate for each mutant. The saturation curves are plotted using data from 3 independent replicates. Error bars represent standard deviation. (TIF)

## Acknowledgments

The authors thank R. Waeckerlin (University of Calgary) for collecting bovine serum samples, and A. Schryvers (University of Calgary) for providing bacterial strains. We thank R. M. Haynes from the Pacific Norwest Cryo-EM Center (PNCC) for assistance and discussions on cryo-EM data collection. We also thank Craig Simpson at SPARC BioCentre, Hospital for Sick Children, Toronto, Canada for performing mass spectrometry data generation. We also thank the Canadian Light Source staff at the Canadian Macromolecular Crystallography Facility for help with data collection. Part of the research described in this paper was performed using beamline CMCF-ID at the Canadian Light Source, and national research facility of the University of Saskatchewan, which is supported by the Canada Foundation for Innovation (CFI), the Natural

Sciences and Engineering Research Council (NSERC), the National Research Council (NRC), the Canadian Institutes of Health Research (CIHR), the Government of Saskatchewan, and the University of Saskatchewan.

## Author contributions

**Conceptualization:** Quynh Huong Nguyen, Chun Heng Royce Lai, Trevor F. Moraes.

**Data curation:** Quynh Huong Nguyen, Chun Heng Royce Lai.

**Formal analysis:** Chun Heng Royce Lai, Michael J. Norris.

**Funding acquisition:** Quynh Huong Nguyen, Trevor F. Moraes.

**Investigation:** Trevor F. Moraes.

**Methodology:** Quynh Huong Nguyen, Chun Heng Royce Lai, Dixon Ng, Christine Chieh-Lin Lai, David E. Isenman, Trevor F. Moraes.

**Project administration:** Quynh Huong Nguyen, Trevor F Moraes.

**Resources:** David E. Isenman, Trevor F. Moraes.

**Supervision:** Trevor F. Moraes.

**Validation:** Quynh Huong Nguyen, Chun Heng Royce Lai, Michael J. Norris, Megha Shah.

**Visualization:** Quynh Huong Nguyen.

**Writing – original draft:** Quynh Huong Nguyen, Trevor F. Moraes.

**Writing – review & editing:** Quynh Huong Nguyen, Chun Heng Royce Lai, Dixon Ng, Megha Shah, Christine Chieh-Lin Lai, David E. Isenman, Trevor F. Moraes.

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
