## [Decision Letter · Decision Letter 0]

9 Dec 2024

PPATHOGENS-D-24-02264

A surface lipoprotein on Pasteurella multocida binds complement factor I to promote immune evasion.

PLOS Pathogens

Dear Dr. Moraes,

Thank you for submitting your manuscript to PLOS Pathogens. After careful consideration, we feel that it has merit but does not fully meet PLOS Pathogens's publication criteria as it currently stands. Therefore, we invite you to submit a revised version of the manuscript that addresses the points raised during the review process.

Please submit your revised manuscript within 60 days Feb 07 2025 11:59PM. If you will need more time than this to complete your revisions, please reply to this message or contact the journal office at plospathogens@plos.org. Please include the following items when submitting your revised manuscript:

We look forward to receiving your revised manuscript.

Kind regards,

Congli Yuan

Academic Editor

PLOS Pathogens

David Skurnik

Section Editor

PLOS Pathogens

 Sumita Bhaduri-McIntosh

Editor-in-Chief

PLOS Pathogens

orcid.org/0000-0003-2946-9497

 Michael Malim

Editor-in-Chief

PLOS Pathogens

orcid.org/0000-0002-7699-2064

**Journal Requirements:**

At this stage, the following Authors/Authors require contributions: Quynh Huong Nguyen, Chun Heng Royce Lai, Michael J Norris, Dixon Ng, Megha Shah, Christine Chieh-Lin Lai, David E Isenman, and Trevor F Moraes. Please ensure that the full contributions of each author are acknowledged in the "Add/Edit/Remove Authors" section of our submission form.

https://journals.plos.org/plospathogens/s/submission-guidelines#loc-parts-of-a-submission

4) We do not publish any copyright or trademark symbols that usually accompany proprietary names, eg ©,  ®, or TM  (e.g. next to drug or reagent names). Therefore please remove all instances of trademark/copyright symbols throughout the text, including:

- TM on Line: 546.

5) Please upload all main figures as separate Figure files in .tif or .eps format. For more information about how to convert and format your figure files please see our guidelines:

6) Please ensure that all Figure files have corresponding citations and legends within the manuscript. Currently, Figure Figure 6 in your submission file inventory does not have an in-text citation. If the figure is no longer to be included as part of the submission, please remove it from the file inventory.

7) We have noticed that you have uploaded Supporting Information files, but you have not included a list of legends. Please add a full list of legends for your Supporting Information files after the references list.

8) We notice that your supplementary Figures, and Tables are included in the manuscript file. Please remove them and upload them with the file type 'Supporting Information'. Please ensure that each Supporting Information file has a legend listed in the manuscript after the references list.

9) Some material included in your submission may be copyrighted. According to PLOSu2019s copyright policy, authors who use figures or other material (e.g., graphics, clipart, maps) from another author or copyright holder must demonstrate or obtain permission to publish this material under the Creative Commons Attribution 4.0 International (CC BY 4.0) License used by PLOS journals. Please closely review the details of PLOSu2019s copyright requirements here: PLOS Licenses and Copyright. If you need to request permissions from a copyright holder, you may use PLOS's Copyright Content Permission form.

Potential Copyright Issues:

- Figures: 4C, and 5; Please confirm whether you drew the images / clip-art within the figure panels by hand. If you did not draw the images, please provide (a) a link to the source of the images or icons and their license / terms of use; or (b) written permission from the copyright holder to publish the images or icons under our CC BY 4.0 license. Alternatively, you may replace the images with open source alternatives. See these open source resources you may use to replace images / clip-art:

10) Please amend your detailed Financial Disclosure statement. This is published with the article. It must therefore be completed in full sentences and contain the exact wording you wish to be published. Please ensure that the funders and grant numbers match between the Financial Disclosure field and the Funding Information tab in your submission form. Note that the funders must be provided in the same order in both places as well. State the initials, alongside each funding source, of each author to receive each grant. For example: "This work was supported by the National Institutes of Health (####### to AM; ###### to CJ) and the National Science Foundation (###### to AM)." State what role the funders took in the study. If the funders had no role in your study, please state: "The funders had no role in study design, data collection and analysis, decision to publish, or preparation of the manuscript.".

**Reviewers' Comments:**

Reviewer's Responses to Questions

**Part I - Summary**

Reviewer #1: The results presented are impressive and obtained with a large array of complementary techniques and the experimental approach is sound. The mechanistic conclusions are justified considering the data presented, and the discussion covers most required topics. Overall, the manuscript represents an important contribution to the field of complement biology, and especially to our understanding of the molecular mechanisms behind pathogen complement evasion. The manuscript deserves publication after revision that considers the comments below.

Reviewer #2: Summary: This manuscript describes characterization of a novel complement evasion mechanism deployed by the livestock pathogen, Pasteurella multocida. Per the authors’ description, P. multocida is a Gram-negative zoonotic pathogen that causes severe diseases associated with high rates of mortality in cattle, including bovine respiratory disease and hemorrhagic septicemia, as well as wound infections in humans. Owing to their recent studies toward P. multocida vaccine development, the authors interest has turned toward an outer membrane protein known as PmSLP-1. A recent paper from the same group (Islam et al, PLoS Pathogens, 2023) has shown that PmSLP-1 is an effective vaccine antigen that provides protective immunity in cattle. However, the function of PmSLP-1 remains unknown. This is the subject of the authors’ present investigation.

The authors begin by using a heterologous E. coli expression system to show that PmSLP-1 is exposed on the bacterial surface. This experiment provides some basic architectural information on the PmSLP-1 protein, which is bolstered by solving a high-resolution crystal structure of a large fragment of the PmSLP-1 extracellular region (residues 95-318). Thereafter, the authors “go fishing” for potential PmSLP-1 binding partners in bovine serum, whereby they identify a nanomolar-affinity interaction with complement Factor I (FI). This interaction is mediated by the structured region of PmSLP-1 (i.e. residues 95-318). It also occurs with FI from sheep and goats, which are two other potential hosts of P. multocida. The authors then go on to define the structure of the PmSLP-1/FI complex using cryo-EM methods. They subsequently validate their structural observations by characterizing a relatively large panel of site-directed mutants in PmSLP-1 and by using amine-reactive crosslinking to define contacts between the two binding partners; the results of these two sets of experiments are internally consistent with the PmSLP-1/FI cryo-EM structure. In the last section of the results, the authors show that the PmSLP-1/FI complex alone is sufficient to promote cleavage of C3b and C4b (using human C3b and C4b as surrogates for the bovine protein). They conclude their experiments by demonstrating that heterologous expression of PmSLP-1 in E. coli allows for bacterial growth in diluted, complement-preserved bovine serum, so long as the PmSLP-1 protein is correctly localized to the cell surface. This demonstrates that recruitment of FI to the bacterial surface by PmSLP-1 provides a measure of protection against complement-mediated killing. Thus, the authors propose that PmSLP-1 be renamed “FI-Binding Protein (fibp)”.

Evaluation: This is a well-conceived and well-executed study that primarily employs a biochemical approach to define a novel complement evasion mechanism. The manuscript was a pleasure to read, and placed the emerging results from the current study within an appropriate context alongside the existing literature in the field. The observations presented should be of interest to those working toward vaccine development against P. multocida and related organisms, but also members of the complement community. The structural data and biochemical data are convincing on their own. However, demonstrating direct recruitment of the FI protease to its substrates (i.e. C3b and C4b) in the absence of known regulatory molecules, such as CR1, FH, and C4BP, is novel to my knowledge and therefore quite exciting.

An additional point that is likely to go unappreciated by many concerns the organization of the FI active site, as described in lines 253-275. FI - like many proteases within the complement system - requires binding to exosites contributed by either a cofactor molecule or regions of the substrate distal to where proteolytic cleavage itself occurs – to assemble an enzymatically-active catalytic triad. The authors’ observations with the covalent inhibitor, PMSF, as well as from their PmSLP-1/FI cryo-EM structure (see Fig. 4f) show that PmSLP-1 binding is sufficient to trigger those changes needed to result in FI enzymatic activity. Thus, PmSLP-1 could be a useful tool for subsequent biochemical studies of FI that may not have been practical otherwise.

Despite my obvious enthusiasm for this manuscript, there are a few small typographical errors that should be corrected prior to acceptance. I have also identified three more substantive concerns that should be addressed. While I’ve listed all of these below, the third point is the most important: an additional well-defined set of experiments is needed to fill the lone gap in this story.

Reviewer #3: Pasteurella multocida is an important pathogen and apart from its zoonotic potential affecting cattle and has a significant economic impact, so understanding of the mechanisms of its pathology is highly relevant and of broad interest.

The present manuscript by Nguyen et al., follows on a pioneering work on the same topic by the same group - (Islam et al., 2023 https://doi.org/10.1371/journal.ppat.1011249), reporting on the isolation of the novel surface lipoprotein PmSLP and its different isoforms.

Here, the study is extended to the structure of the dominant isoform PmSLP-1 both in isolation and in complex with complement factor I, which is identified to be an effector of the PmSLP-1. The latter cryo-EM structure also elucidates the mechanism of FI stimulation by the PmSLP by stabilizing the catalytic domain.

The finding is significant not only as it rationalizes the action of the PmSLP, which appears to be a promising vaccine candidate, but also reports for the first time the direct activation of complement factor I by a bacterial protein.

The protein PmSLP is delivered by the recently characterized Type XI secretion system, and is transported across the outer membrane by a dedicated Slam protein. As such this study also provides major leap in understanding of this poorly studied secretion system that is widely spread across the proteobacteria.

The work is well-executed and with its broad, interdisciplinary scope and significant novelty, it merits publication in the PloS Pathogens with minor corrections.

**Part II – Major Issues: Key Experiments Required for Acceptance**

Reviewer #1: Authors determine the structure of a two-fold symmetric dimer FI:PmSLP-1 complex by cryo-EM. The dimerization appears to be through FI. Discuss whether the dimeric complex is in vivo relevant, and can FI access the C3b substrate in the dimeric state? Is it expected that the dimer form on the surface of Pasteurella multocida?

Multiple panels with detailed views need to be improved considerably, see specific comments.

Reviewer #2: Major Points:

1/ Figures, Fig. 3, 4, Ext4, Ext5, and Ext7: Although I’m not familiar with Journal style guides, the figures mentioned here are problematic because they extend across more than one printed page. I’m sure things will be rearranged and scaled accordingly (if accepted for publication), but it’s confusing and inconvenient to have to flip pages to evaluate information contained within a single figure itself.

2/ BLI Studies: The authors present a large number of BLI experiments to determine affinities of wild-type PmSLP-1 and various mutants for bovine FI. While these experiments themselves are not problematic, it seems that the authors chose only to use dose-response analysis to evaluate their data. However, there ought to be kinetic information, such as on-rates and off-rates, obtainable from these experiments. At a minimum, it would be good to show these kinetic rate constants for the wild-type protein. The mutants can be compared to wild-type by dose-response analysis, as is already done, for the sake of simplicity.

3/ PmSLP-1/FI binding to C3b and C4b: The greatest weakness in this study the lack of direct binding data between the PmSLP-1/FI complex and the target molecules, C3b and C4b. To their credit, the authors point this out in the Results (lines 327-330), wherein they suggest that the interaction must be transient as neither C3b nor C4b was observed in their pulldown experiment. While I agree that bands corresponding to either C3b or C4b were not found in the pulldown (see Fig. 2a, b), it’s also true that blood/plasma/sera samples are often collected in the presence of agents like EDTA, citrate, etc., that inhibit complement activation; unfortunately, the experimental section doesn’t say much about how the serum was obtained, other than the fact that it was taken from healthy donor cattle. Nevertheless, there should be negligible C3b and C4b in the samples used for the pulldown because a specific complement activator was likely not present in the first place.

Independent of any potential problems with their explanation outlined above or how the serum was obtained, I think the authors should investigate the affinity of the complex they discovered here - as well as that of PmSLP-1 itself - for bovine C3b and C4b. I note that the authors used biotinylated proteins as ligands for their BLI studies. In that regard, there exist numerous protocols for labeling C3b or C4b at the thioester cysteine exposed upon activation of C3 and C4. Purification of C3 and C4 from serum is straightforward and has been done for decades, and bovine serum is plentiful. So, attempting to address this weakness wouldn’t necessarily mean “reinventing the wheel”, so-to-speak. On the other hand, addressing this issue will fill the single most obvious gap in the results of this manuscript, which is otherwise very strong.

Reviewer #3: As mentioned above, this is a thorough and well-executed piece of work, which doesn't appear to require any major additional experimental validation to support its central claims. It also has sufficient novelty and impact to justify its appearance in PloS Pathogens.

**Part III – Minor Issues: Editorial and Data Presentation Modifications**

Reviewer #1: Minor comments

Grammar needs to be checked throughout. In many places “the” is missing.

General note on figures. In many of the close-up views, the perception of depth can be improved by reducing the clip size (distance between the two planes between which things are visible) in the direction perpendicular to the screen. Several such figures would benefit from clip optimization. One example is the close-up view in ext data Fig 5d.

Fig 1C should be improved and expanded considerably. Color the domains differently and present details of the structure. Include a panel illustrating the quality of the electron density, e.g. from the domain interface. Are the two domains stably associated; what type of interactions are keeping the domains together? The expanded view with the three lysine mutations, why is this important for the reader?

Ext table 1. Provide statistics for Ramachandran and clash score

Line 123. Improve language

Line 157. SEC is 2C not 2D

Ext Fig 3C-D. Please superimpose the fitted curves on the measured association and dissociation curves. Is a 1:1 binding model assumed?

Line 188. How close was the AF (alphafold) predicted model to the experimental structure?

Line 190. The interface is given as 1374 Å2 (is it an area, so Å2, not Å). Since this means that you suggest the uncertainty is 1 Å2 then why use the word approximately? And this is not buried surface area instead of interface area? Is the area from PISA analysis

Line 193-194. Please describe the hydrophobic interactions, provide a figure presenting these hydrophobic interactions

196-197. It seems like Ext fig 2 should be moved, it is much more meaningful after having presented the structure.

Line 203. It should be stressed that PmSLP-1 and FH do not share structural homology. It is not correct to state that they interact in the same manner, instead they recognize the same patch on FI, which is something different. Could the FH-PmSLP-1 overlap be quantitated in some way?

Line 207. The overlap w MG6 in C3b is interesting. Any thoughts on how this could be overcome or whether perhaps pMSLP-1 does not need to form specific interactions with C3b? Could there be conformational changes in C3b or PmSLP-1 that could allow formation of ternary complex. Have authors attempted to measure formation of a ternary complex experimentally?

Line 213. Specify which complex.

Line 236. This is important and impressive data, but the location of the mapped residues and their interaction partners in FI across the intermolecular interface deserves to be presented much better. The overall cartoon based location in fig 3e is not that informative in this respect.

Line 330. Bovine C3 is easy to purify and bovine C3b can be obtained with mild trypsin treatment. It would strengthen the manuscript considerably if the BLI experiment included also bovine C3b. One could also immobilize bovine C3b through thioester Cys-Maleimide-biotin to sensor and attempt a build-up with the preassembled PmSLP-1:FI complex. This is a standard approach for human C3b interaction measurements. The fact that the complex is not isolated in pulldown does not exclude an interaction with Kd in the low micromolar range

Line 346. The FI interaction with the CTC may not be mandatory, e.g. see PMID: 32938727. Differences between the rotation of CTC may be influenced by crystal packing.

Line 351. FI SP is not highly disordered. Regions may be dynamic in free FI compared to FH:C3b:FI complex. Disordered has a special meaning in structural biology which is different from the manner it is used here.

Fig 5. There is no need to duplicate for AP and CP/LP , the principle is the same

Line 404. Insert consistently space between number and unit. One example where this not done is 0.45uM here. Check this throughout.

Line 409. “When removal of the carboxyl terminal His-tag is needed” Stick to past time.

Line 414. What buffer was used for the SEC purification? Is the column perhaps from Cytiva?

Line 420.Mutation to Ala?

Line 433. Screened for crystallization.

Line 438. Reservoir condition supplemented with 20% v/v glycerol?

Line 445. Data revealed an anomalous signal extending to 2 Å. How was this evaluated? Since the diffraction limit is also 2 Å, it is unlikely that the anomalous difference is significant at 2 Å also. Such a signal is weak and will normally not be significant in the last 0.5 Å resolution shell of such 2 Å data. Not that it matters for experimental phasing! It would also strengthen the manuscript if statistics for the experimental phasing in PHENIX was included in the manuscript.

Line 479. Where is the Titan microscope located? It is mentioned in the acknowledgements, but state it here also.

Line 501. Can the difference in protomer orientation between the two 3D classes be quantitated? Are the two protomers differing by a simple small rotation or are there also internal differences within the protomers. RMSD values between protomers from the two different 3D classes could be given as a measure here.

Line 568. Purified bovine FI?

Line 595. Correct “35 to 95oC”

Line 613. “Supplementary Table 2” this must be extended table 2?

Line 640. This is Extended data 2, not 4.

Ext fig 2B. Very low resolution, export high-res from Excel

Legend fig 1. “Membrane was stained with ponceau S to show the amount of sample loaded per lane and detect cell lysis” . It is not the membrane that is stained, it must be the proteins from the membrane

Figure 3. More close-up views to document the quality of the map would be beneficial. In b, the mesh thickness should be decreased or a transparent surface used instead for the map. Also provide the contour level everywhere a map is presented. The local resolution map in the Extended figure could be moved here. Panel C, outline the interface footprint on molecules in the right panel. Panel E, how were these residues selected? From the structure only?

Are there any conformational changes between the unbound crystal structure and PmSLP-1 in the complex w FI?

Ext data table 1. How was the table generated? Phenix.table_one? As mentioned above, please include statistics from the experimental phasing on the anomalous data.

Ext data fig 1. The rmsd values given are not that meaningful. Provide the rmsd for the two domains separately and only for the common secondary structure elements.

Ext data fig 4. The figure is so large that it should be split in two. Panel B should come early since it presents the sample rather than in the end

Ext table 3. Coot is not used for refinement, only for rebuilding. Again, what program was used to generate the table? phenix.validation_cryoem?

Ext fig 5. Panel A. Give a reference to PISA , provide the weblink if the EBI server was used. Explain the meaning of P value from PISA and perhaps discuss whether it is a problem that the P value for the complex is close to 0.5. Panel B. The C3b docking is made under the assumption that C3b conformation required for PmSLP-1:FI to cleave is the same as that required for FH:FI cleavage. This is not necessarily true. There are many structures of C3b in the PDB, and the location of the CUB domain relative to the MG-ring in a different known structure may reduce or remove the clash observed here for PmSLP-1 and C3b-MG6. Authors may want to compare with other known C3b containing structures. Also, pMSLP-1 bound FI may not require the CTC contact observed in the FI:FIH.C3b complex.

Ext data fig 6. Panel C. why are the crosslinks mapped onto AF predicted structure. Authors have an experimental structure, that should be trustworthy at the resolution the cryo-EM map. Or is there a specific reason?

Reviewed by Gregers R Andersen

Reviewer #2: Minor Points:

1/ Fig. 2, legend: The word “condition” should be pluralized to “conditions” when describing preparation of samples for SDS-PAGE. Additionally, the legend inset of Fig 2d refers to a construct PmSLP-1^95, while the legend text refers to a construct PmSLP-1^94; please revise this accordingly.

2/ Fig. 4, panels a, b: There does not seem to be a label, arrow, or text indicating the presence of the PmSLP-1 band in these SDS-PAGE images. This should be added to make this experiment easier to interpret for the readers.

3/ Fig. EXD 6, legend: It seems the article “the” is missing from the title prior to “PmSLP-1:FI complex”. Also, in the last sentence of this legend, the word “occurrence” should be pluralized to “occurrences”.

4/ Abstract, line 13: The word “brutal” seems a bit excessive for the Abstract of a scientific manuscript. It might be worth removing this.

5/ Introduction, line 54: The authors suggest that ‘cleaving of C3b and C4b into their inactive form….brings [all activity of] the complement cascade to a halt’. I think this is overstated, because its likely that there are many molecules of C3b and C4b generated upon complement activation and the word “halt” implies a sort of immediacy that’s not likely seen when these natural regulatory mechanisms are in play. This problem can be eliminated by using a different word like “inhibited” or “diminished”, etc.

6/ Results, line 139: The article “the” is missing prior to the word “PmSLP-1 sample”.

7/ Results, line 154: Instead of using a generic word like “tight”, it would be better if the authors used a more scientifically informative description like “a nanomolar-affinity” or “a high-affinity” instead.

8/ Results, lines 192-194: The authors write that “hydrophobic interactions also play a role in stabilizing the complex”. This is not necessary, as it is typical for protein-protein interactions – in fact, it would be very unusual if hydrophobic interactions weren’t involved to some degree. I recommend removing this sentence.

Reviewer #3: General minor comment: Figures 3 and 4 are multipaneled and spill over 2 pages, making it difficult to follow. May be a better idea to split them into several figures.

The crystal structure presented lacks 94 N-terminal residues. It would be helpful to provide the logic of the introduction of the “surface entropy reducing mutations” , or a general overview of the organization of the full-length molecule of PmSLP-1.

The binding of the PmSLP-1 and FI is surprisingly tight, with a nanomolar affinity. The assessment of the docking interface is very diligent, and doesn’t simply rely on the cryo-EM structure or docking simulations, but a number of site-directed mutants are introduced and tested, which is very commendable. It would perhaps be helpful to show the putative interaction partners between E240 and respective FI residues, as the current Fig 3 isn’t very clear and E240 seems buried inside PmSLP-1.

Extended Data Fig 2b – the rendering of the table of the identity matrix is badly compressed and shows artifacts – please ensure that is rectified in the final version.

Extended Data Fig 3a/b – could hydrodynamic radius be calculated and used instead of the retention volume, which is column specific?

Extended Data table 4 – the list of plasmids should perhaps be a bit more detailed in specifying the exact length of the constructs, any linker regions and the cloning sites used. A UniprotKB (or other curated database) ID or the full sequence of the protein should also be provided, as it is for the majority of the other proteins discussed in the text, so one shouldn’t be forced to go via the PDB database to find it.

Methods: please convert rpms to g-forces where applicable. Also, in some cases e.g. line 592 – formatting of the centigrade is missing.

PLOS authors have the option to publish the peer review history of their article (what does this mean? ). If published, this will include your full peer review and any attached files.

**Do you want your identity to be public for this peer review?** For information about this choice, including consent withdrawal, please see our Privacy Policy .

Reviewer #1: **Yes: ** Gregers Rom Andersen

Reviewer #2: No

Reviewer #3: No

**Figure resubmission:**
---

## [Decision Letter · Decision Letter 1]

27 Mar 2025

Dear Dr. Moraes,

We are pleased to inform you that your manuscript 'A surface lipoprotein on Pasteurella multocida binds complement factor I to promote immune evasion.' has been provisionally accepted for publication in PLOS Pathogens.

Best regards,

Congli Yuan

Academic Editor

PLOS Pathogens

David Skurnik

Section Editor

PLOS Pathogens

Sumita Bhaduri-McIntosh

Editor-in-Chief

PLOS Pathogens

orcid.org/0000-0003-2946-9497

Michael Malim

Editor-in-Chief

PLOS Pathogens

orcid.org/0000-0002-7699-2064

Reviewer Comments (if any, and for reference):

Reviewer's Responses to Questions

**Part I - Summary**

Reviewer #1: Revised manuscript much improved

Reviewer #2: The authors have significantly revised their manuscript, especially as far as figure presentation is concerned. Although additional experiments were requested, the results were either negative or the data obtained are reserved for inclusion in a follow-on study. This approach is justified, as this revised manuscript as presented tells a complete story on its own. In my opinion, the manuscript should be accepted for publication.

Reviewer #3: The revision addresses all of the points raised associated with the original submission and the additional binding assays performed with PmSLP-1 homologues binding to C3b and C4b also address some of the points raised by Reviewer 2, although provision of these as a supplemental figure would have been appreciated.

Pverall, I am satisfied with the current state of the manuscript and I trust it is suitable for publication in its current form.

**Part II – Major Issues: Key Experiments Required for Acceptance**

Reviewer #1: Not applicable

Reviewer #2: n/a

Reviewer #3: The revision addresses most of the points raised associated with the original submission and I have no reservation in recommending it for publication.

**Part III – Minor Issues: Editorial and Data Presentation Modifications**

Reviewer #1: Fig 1C, right panel. If selected residues were labeled to allow the reader to orient, it would improve the figure

Figure 2. Legend to panel G is presented as legend to panel E in the end

Table S2. Hydrogen bonds (HBs) longer than 3.3 are not strong. Consider to remove such HBs from the table or at least verify that the angle is not less than say 135 deg for such long HBs

In the text, reference to supporting figures and tables are of mixed style e.g. Line 179 ”Table S3”, Line 155 ”S1 Fig”. This should be made consistent with journal style.

Table S3. #10 The anaphylatoxin C4a does not have Mw of 192 kDa. The entire C4 has this Mw. Please correct what was actually identified, C4a or C4

Table S4. ColabFold2. This is perhaps Alphafold2 running on Colab?

Table S4. Clashscore is high for a 2025 structure. Consider repeating the last refinement in phenix.realspace_cryoEM with default nonbonded_weight = 100 in the range 1000-3000, that should reduce the clash score to < 3 quickly. If this is done, the interaction analysis should be redone on the final structure.

S3 Fig panel A-B. The fractions analyzed by SDS-PAGE are probably from peaks labeled Peak 1 and Peak 3 on the chromatogram. This figure is redundant with Figure S4A-B.

S4 Fig A-B, shade the area on A where the analyzed fractions come from. The fraction labels B9 etc in panel B are not informative

Line 366. The distribution of the key residues (bolded) on the surface 367 of PmSLP-1 could also be seen in Fig 4B (right panel). Strange sentence, revise

Reviewer #2: n/a

Reviewer #3: All the issues identified in the original submission have been rectified.

PLOS authors have the option to publish the peer review history of their article (what does this mean? ). If published, this will include your full peer review and any attached files.

**Do you want your identity to be public for this peer review?** For information about this choice, including consent withdrawal, please see our Privacy Policy .

Reviewer #1: **Yes: ** Gregers Rom Andersen

Reviewer #2: No

Reviewer #3: No

---

## [Editor Report · Acceptance letter]

Dear Dr. Moraes,

We are delighted to inform you that your manuscript, "A surface lipoprotein on Pasteurella multocida binds complement factor I to promote immune evasion.," has been formally accepted for publication in PLOS Pathogens.

Best regards,

Sumita Bhaduri-McIntosh

Editor-in-Chief

PLOS Pathogens

orcid.org/0000-0003-2946-9497

Michael Malim

Editor-in-Chief

PLOS Pathogens

orcid.org/0000-0002-7699-2064